# Targeting Nup358/RanBP2 by a viral protein disrupts stress granule formation

Jibin Sadasivan[1], Marli Vlok[1], Xinying Wang[1], Arabinda Nayak[2], Raul Andino[2], Eric Jan[1]*

**1** Department of Biochemistry and Molecular Biology, Life Sciences Institute, University of British Columbia, Vancouver, British Columbia, Canada, **2** Department of Microbiology and Immunology, University of California San Francisco, San Francisco, California, United States of America

* ej@mail.ubc.ca

**Data Availability Statement:** All relevant data are within the manuscript and its Supporting Information files.

**Funding:** This study was supported by Canadian Institute of Health Research operating grant (PJT-

## Abstract

Viruses have evolved mechanisms to modulate cellular pathways to facilitate infection. One such pathway is the formation of stress granules (SG), which are ribonucleoprotein complexes that assemble during translation inhibition following cellular stress. Inhibition of SG assembly has been observed under numerous virus infections across species, suggesting a conserved fundamental viral strategy. However, the significance of SG modulation during virus infection is not fully understood. The 1A protein encoded by the model dicistrovirus, *Cricket paralysis virus* (CrPV), is a multifunctional protein that can bind to and degrade Ago-2 in an E3 ubiquitin ligase-dependent manner to block the antiviral RNA interference pathway and inhibit SG formation. Moreover, the R146 residue of 1A is necessary for SG inhibition and CrPV infection in both Drosophila S2 cells and adult flies. Here, we uncoupled CrPV-1A's functions and provide insight into its underlying mechanism for SG inhibition. CrPV-1A mediated inhibition of SGs requires the E3 ubiquitin-ligase binding domain and the R146 residue, but not the Ago-2 binding domain. Wild-type but not mutant CrPV-1A R146A localizes to the nuclear membrane which correlates with nuclear enrichment of poly(A)+ RNA. Transcriptome changes in CrPV-infected cells are dependent on the R146 residue. Finally, Nup358/RanBP2 is targeted and degraded in CrPV-infected cells in an R146-dependent manner and the depletion of Nup358 blocks SG formation. We propose that CrPV utilizes a multiprong strategy whereby the CrPV-1A protein interferes with a nuclear event that contributes to SG inhibition in order to promote infection.

## Author summary

Viruses often inhibit a cellular stress response that leads to the accumulation of RNA and protein condensates called stress granules. How this occurs and why this would benefit virus infection are not fully understood. Here, we reveal a viral protein that can block stress granules and identify a key amino acid residue in the protein that inactivates this function. We demonstrate that this viral protein has multiple functions to modulate nuclear events including mRNA export and transcription to regulate stress granule formation. We identify a key host protein that is important for viral protein-mediated stress

178342) https://cihr-irsc.gc.ca/e/193.html to EJ; Natural Sciences and Engineering Research Council of Canada Discovery grant (RGPIN-2017-04515) https://www.nserc-crsng.gc.ca/index_eng.asp to EJ; National Institutes of Health (A132131 - R01AI137471) https://www.nih.gov/ to RA and SERB-UBC Doctoral scholarship to JS. The funders had no role in study design, data collection and analysis, decision to publish, or preparation of the manuscript.

granule inhibition, thus providing mechanistic insights. This study reveals a novel viral strategy in modulating stress granule formation to promote virus infection.

## Introduction

Stress granules (SGs) are dynamic, non-membranous, cytosolic aggregates of ribonucleoprotein (RNP) complexes that assemble following cellular stress [1]. Typically, overall translational inhibition resulting from a cellular stress response promotes SG formation, but is not necessary under certain cellular contexts [2–4]. SGs contain non-translating mRNAs, translation initiation factors, and ribonucleoproteins [5]. The assembly of SGs is mediated through the interactions of proteins with non-translating RNAs, resulting in liquid-liquid phase separation, where in part the RNA component serves as scaffolds for recruitment of RNA binding proteins. SG assembly is proposed to be a multistep process in which the assembly of a stable dense core of mRNA and proteins is held together by a surrounding shell of less concentrated RNPs [6]. Common SG protein markers include RasGAP-SH3-binding protein (G3BP1), T-cell intracellular antigen 1 (TIA-1), TIA-1 related protein (TIAR), and Poly-A binding protein (PABP), however, hundreds of other proteins have been identified that are enriched in SGs [7,8]. Moreover, relatively long mRNAs are enriched in SGs, possibly to promote concentration of proteins for liquid-liquid phase separation [9,10]. SGs are dynamic and reversible structures that continuously sort and route messenger RNP (mRNP) components. SGs affect mRNP localization, functions and signaling pathways that can have significant impacts on biological processes [11]. As such, the dysregulation of SG assembly/disassembly is implicated in neurodegenerative diseases, autoimmune diseases, cancers and virus infections [12]. Over the past decade, significant progress has been made in unraveling the SG composition and assembly pathways. However, the molecular mechanism and underlying signaling pathways that regulate SG dynamics, and the consequences of SG assembly are not completely understood.

Classical induction of SG assembly is initiated by the activation of one or more stress-sensing eIF2α kinases, that phosphorylate serine-51 of the α subunit of eukaryotic translation initiation factor 2 (eIF2), which is the main factor that delivers initiator Met-tRNA to the 40S pre-initiation complex [13]. In mammals, there are four eIF2α kinases, Protein kinase R (PKR), Protein kinase RNA-like endoplasmic reticulum kinase (PERK), Gene control nonderepressible 2 (GCN2) and Heme-regulated inhibitor kinase (HRI) [14–17]; whereas in insects, there are only two, PERK and GCN2 [18]. Phosphorylation of eIF2α results in inhibition of overall translation in the cell which can lead to robust SG formation. As a result, besides hallmark SG protein markers and poly(A)+ RNA, several eukaryotic translation initiation factors and the 40S subunit are often found in SG foci [19]. Although it is often thought that translation inhibition is a pre-requisite for SG formation, this is not strictly necessary under certain cellular contexts [20].

Virus infection, in general, leads to modulation and inhibition of SG formation [21,22], which is observed across different classes of RNA and DNA viruses and across species suggesting a fundamental viral strategy to modulate SGs for productive infection. For example, RNA viruses such as poliovirus and hepatitis c virus (HCV) infection leads to a depletion of G3BP1 and TIA-1 foci formation [23,24]. The disruption of SG assembly can be attributed to one or more viral proteins, which has revealed distinct mechanisms that affect SG. One such mechanism is to counter SG assembly by modulating PKR activation. For instance, Middle East Respiratory Syndrome (MERS) Coronavirus accessory protein 4a, Influenza virus NS1 protein and Vaccinia virus E3L inhibits SG formation by sequestering dsRNA to block PKR activation

[25–28]. Kaposi's sarcoma-associated herpesvirus (KSHV) ORF57 protein binds to PKR and PKR activating protein (PACT) to inhibit PKR activation and SG formation [29]. Besides modulating PKR activity, some viruses act directly on SG through virally-encoded proteases that cleave key SG proteins to facilitate SG disassembly. Poliovirus 3C protease and Foot-and-mouth disease virus (FMDV) 3C and Leader proteases cleave G3BP to inhibit SG formation [23,30,31]. Viruses also co-opt SG components to facilitate infection. Flaviviruses such as West Nile virus and Zika virus hijack SG-nucleating proteins TIA-1,TIAR and G3BP and subvert them to viral replication complexes [32,33] whereas Human Immunodeficiency virus-1 (HIV-1) sequesters the SG protein Staufen-1 to RNPs containing viral RNA and gag protein [34]. Murine norovirus utilizes the NS3 protein to redistribute G3BP to the site of viral replication [35]. In addition, the Severe Acute Respiratory Syndrome-Coronavirus-2 (SARS-CoV-2) nucleocapsid protein phase separates with G3BPs and rewires the G3BP interactome to disassemble SGs [36–38]. The distinct mechanisms and utilization of viral proteins to disassemble SGs across different virus classes highlight the importance of SG modulation during virus infection.

Although it is apparent that viruses modulate SGs, the reasons underlying this event are not fully understood. SG formation may sequester viral protein or RNA, as observed with flavivirus infection [39], thus inhibition of SG may be a general viral strategy to allow viral protein synthesis and replication. Alternatively, antiviral RNA sensors and factors such as PKR, Retinoic acid inducible gene I (RIG-I), Melanoma differentiation-associated protein 5 (MDA5), oligoadenylate synthetase (OAS), ribonuclease L (RNase L), Tripartite motif containing 5 (Trim5), RNA-specific adenosine deaminase 1 (ADAR1) and cyclic GMP-AMP synthase (cGAS) have been found in SGs, termed antiviral SGs (avSGs), which may act as an antiviral hub to co-ordinate immune responses to limit viral replication [40–43]. Influenza A virus RNA and RIG-1 have been found in avSGs during infection, which is thought to trigger the RIG-I-dependent interferon response [42]. Finally, studies have implicated SG formation in apoptosis, thus blocking SG during infection may delay this process to allow completion of the viral life cycle [44]. The functional consequences of SG formation and its causal relationship to virus infection remains to be clarified.

Dicistroviruses are single stranded positive sense RNA viruses that primarily infect arthropods [45,46]. Members of the family *Dicistroviridae* include the honeybee dicistroviruses, *Israeli acute paralysis virus*, *Kashmiri bee virus* and *Black queen cell virus*, that have been linked to honeybee disease, and *Taura syndrome virus*, which has led to panaeid shrimp outbreaks [47]. The dicistrovirus RNA genome consists of two main open reading frames (ORF) (Fig 1A). ORF1 encodes the viral non-structural proteins, such as the RNA helicase, protease and RNA-dependent RNA polymerase and ORF2 encodes the viral structural proteins, which mediate virion assembly [45]. Both ORFs are driven by distinct internal ribosome entry sites (IRES) that have been studied extensively [48–51]. The intergenic IRES utilizes a streamlined translation initiation mechanism whereby the IRES mediate direct assembly of ribosomes and starts translation at a non-AUG codon [45,52]. The dicistrovirus *Cricket paralysis virus* (CrPV) and *Rhopalosiphum padi virus* (RhPV) 5'UTR IRES resembles an IRES similar to the mechanism used by HCV, requiring translation initiation factors, eIF2, eIF3 and initiator Met-tRNA$_i$ to start translation [53–55]. Studies using model dicistroviruses CrPV and *Drosophila C virus* (DCV) have uncovered fundamental virus host interactions in insects. Dicistrovirus infections can lead to transcriptional and translational shutdown, evasion of the insect antiviral RNAi response and SG inhibition [56–59].

The CrPV and DCV 1A proteins are viral suppressors of RNAi (VSR) that suppress the insect antiviral RNAi pathway [58,59]. The 1A protein is the first viral non-structural protein translated within ORF1. Immediately downstream of the 1A protein is a 2A peptide, which

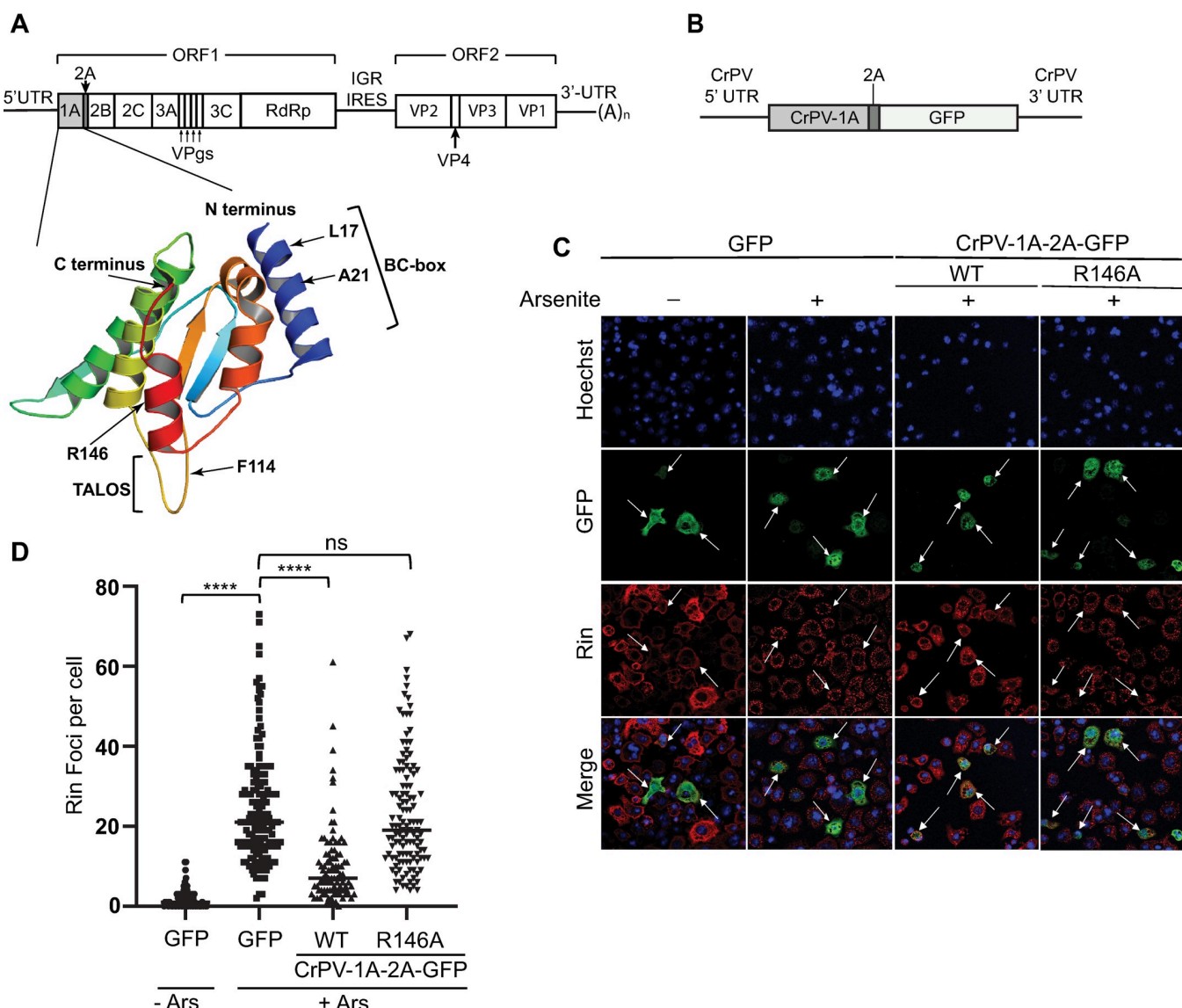

**Fig 1. CrPV-1A expression inhibits stress granules in response to arsenite treatment.** (A) Depiction of the CrPV genome with the structure of CrPV-1A protein (PDB 6C3R) (*below*) highlighting the domains selected for mutagenesis. (B) Schematic of CrPV-1A-2A-GFP RNA containing the CrPV 5' and 3'UTRs. (C) Confocal immunofluorescence images of S2 cells transfected with control 5'cap-GFP-poly (A)+, wild type or R146A mutant CrPV-1A-2A-GFP RNAs (16 hours) followed by one-hour treatment in the presence or absence of 500 μM sodium arsenite. The arrows show transfected cells. Shown are representative transfected cells detecting GFP fluorescence (green), Rin antibody staining (red), Hoechst dye staining for nucleus (blue) and merged images. Images were taken using the Leica Sp5 confocal microscope with a 63X objective lens and 2X zoom (D) Box plot showing the number Rin foci per cell. At least 50 cells were counted for each condition from three independent experiments. Data are mean ± SD. P > 0.05 (ns) p < 0.0001(****) by a one-way ANOVA (nonparametric) with a Bonferroni's post hoc-test.

mediates a "stop-go" translation mechanism that leads to release of the mature 1A protein [60]. DCV-1A, a 99 amino acid protein, is a double-stranded RNA (dsRNA) binding protein that sequesters dsRNA intermediates from Dicer-2 mediated processing by the RNAi machinery. CrPV-1A, a 166 amino acid protein, employs a dual mechanism by which it binds to and inhibits Argonaute-2 (Ago-2) activity and stability [58,59,61]. Ago-2 mutant Drosophila are more susceptible to dicistrovirus infection, demonstrating the importance of the antiviral effects of Ago-2 [61,62]. CrPV-1A binding to Ago-2 inhibits its activity and also leads to Ago-2 degradation via an E3 ubiquitin ligase-dependent pathway [59]. Biochemical and single

molecule studies showed that CrPV-1A inhibits the initial seed base-pairing targeting by Ago-2-RISC (RNA induced silencing complex) [63]. Structural and biochemical analyses have mapped distinct functions to specific domains on CrPV-1A. Specifically, CrPV-1A interacts with Ago-2 through a flexible loop containing a TALOS (targeting argonaute for loss of silencing) element and recruits the host ubiquitin complex, Cul2-Rbx1-EloBC through a BC box domain (Fig 1A) [59]. The F114 residue within TALOS is critical for Ago-2 binding and the L17 and A21 residues in the BC box domain are required for recruitment of the ubiquitin ligase complex [59]. We previously showed that the CrPV-1A protein inhibits SG foci formation and transcription [64]. CrPV-1A's ability to inhibit SG and transcription is mapped to a single R146 residue at the C terminus. Mutant CrPV (R146A) virus infection is attenuated which is correlated with an increase in SG formation, strongly implicating potential antiviral properties of SG formation. Moreover, blocking transcription inhibited SG formation and restored CrPV (R146A) virus infection, suggesting that the SG modulation is linked to a nuclear event(s) [64]. In summary, CrPV-1A is a multifunctional protein that modulates several host cell processes to promote infection. Whether the specific functions of CrPV-1A are mutually exclusive or interdependent have yet to be examined.

In this study, we use overexpression and mutagenesis approaches to uncouple the relationship between the multiple functions of CrPV-1A. We show that CrPV-1A's ability to inhibit SGs is dependent on the BC Box ubiquitin complex-interacting domain and independent of the Ago-2 binding TALOS element. We also demonstrate that CrPV-1A localizes to the nuclear periphery which correlates with nuclear poly(A)+ RNA enrichment. Transcriptome analysis and gene depletion studies suggest that CrPV-1A modulates host steady state RNA levels and mRNA export. Finally, productive CrPV infection requires the nuclear pore complex protein Nup358/RanBP2 in a CrPV-1A R146-dependent manner. We propose that CrPV-1A mediated SG inhibition is linked to nuclear events including transcriptional shutoff and nuclear mRNA accumulation to promote infection.

## Results

### CrPV-1A mediated stress granule inhibition in arsenite-treated cells

SG assembly can be induced through distinct pathways by targeting the activity of specific translation initiation factors [2,65,66]. The CrPV-1A protein is relatively small (166 amino acids) and has multiple functions including inhibition of SG and RNAi [58,59,64]. Mutation of F114 to alanine (F114A) disrupts CrPV-1A interactions with Ago-2 and mutations L17A and A21D within the BC box domain block CrPV-1A recruitment with the Cul2-Rbx1-EloBC complex (Fig 1A). We previously showed that expression of CrPV-1A inhibits SG formation in Drosophila S2 cells treated with pateamine A (Pat A), which is a compound that dysregulates the helicase activity of the translation initiation factor, eIF4A [64,67]. We examined whether inhibition of SGs by CrPV-1A can occur through another stress-induced pathway. Arsenite inhibits global translation by activating the two eIF2α kinases in Drosophila, PERK and GCN2, to inhibit eIF2 activity [18].

To monitor cells that express CrPV-1A, we generated a novel GFP-based mRNA reporter (CrPV-1A-2A-GFP; Fig 1B) containing the CrPV-1A open reading frame (amino acids 1–166; Q9IJX4) fused in frame with GFP and the natural genome arrangement containing the CrPV-2A peptide, which is upstream of GFP thus allowing the "stop-go" translation mechanism to separate the CrPV-1A and GFP proteins [60]. We appended the CrPV 5'-UTR IRES and 3'UTR to ensure expression of CrPV-1A. We transfected *in vitro* transcribed CrPV-1A-2A-GFP RNAs into S2 cells in order to bypass the inhibitory effects of CrPV-1A on transcription [64].

To examine SG assembly, we monitored Rasputin (Rin) foci formation by immunofluorescence in S2 cells transfected with *in vitro* transcribed CrPV-1A-2A-GFP RNA or 5'cap-GFP-poly(A) RNA. Rin is the Drosophila homolog of mammalian G3BP1, which is a hallmark SG marker protein [68]. We previously showed that expression of CrPV-1A resulted in SG inhibition, specifically reducing the number of Rin foci per cell [64]. In cells expressing control GFP, Rin protein remained diffuse in the cytoplasm (Fig 1C). Arsenite treatment of S2 cells transfected with control GFP RNA resulted in robust induction of Rin foci per cell (Fig 1C and 1D). By contrast, cells transfected with the CrPV-1A-2A-GFP RNA resulted in fewer Rin foci per cell in the presence of arsenite treatment, similar to that observed previously under pateamine A treatment [64]. The residue R146 of CrPV-1A is required for CrPV-1A-mediated SG inhibition [64]. Transfection of mutant CrPV-1A(R146A)-2A-GFP in arsenite-treated cells did not reduce the number of Rin foci per cell as compared to cells expressing GFP alone (Fig 1C and 1D). In summary, these results indicate that CrPV-1A expression can inhibit distinct SG assembly pathways and the R146 residue is critical for CrPV-1A mediated SG inhibition.

## Uncoupling CrPV-1A multifunctional domains and stress granule inhibition

To determine whether the effects of CrPV-1A on SG inhibition are associated with other functions of CrPV-1A such as Ago-2 binding, we generated specific or combinations of mutations within the CrPV-1A reporter RNA (Fig 1A). Specifically, we expressed mutant CrPV-1A protein containing either F114A or double mutants F114A/R146A and monitored Rin foci formation in S2 cells under arsenite treatment. Expression of CrPV-1A(F114A)-2A-GFP reduced the number of Rin foci per cell, similar to that observed when wild-type CrPV-1A is expressed (Fig 2A and 2B). By contrast, expression of the double mutant CrPV-1A(F114A/R146A)-2A-GFP did not reduce the number of Rin foci per cell, which is similar to that observed when CrPV-1A(R146A)-2A-GFP is expressed (Fig 2A and 2B). These results strongly showed that CrPV-1A-mediated SG inhibition is independent of Ago-2 binding.

We next investigated whether the BC box domain of CrPV-1A is required for SG inhibition. The BC box domain recruits the Cul2-Rbx1-EloBC complex. Mutations L17A and A21D within the BC box domain block the recruitment of the Cul2-Rbx1-EloBC complex, thereby inhibiting E3 ubiquitin ligase activity [59]. Expression of mutant CrPV-1A containing L17A or A21D mutations did not reduce the number of Rin foci per cell in cells treated with arsenite (Fig 2A and 2B), thus suggesting that the BC box domain is required for inhibition of SG by CrPV-1A. Expression of a double mutant CrPV-1A(R146A/A21D)-2A-GFP in arsenite-treated S2 cells showed similar inhibition to that of single mutant CrPV-1A(R146A)-2A-GFP, thus supporting the conclusion that R146 is required for CrPV-1A's ability to block SG formation. We also investigated the double mutant CrPV-1A(F114A/A21D)-2A-GFP; expression of this mutant led to a similar number of Rin foci per cell as the single mutant CrPV-1A(A21D)-2A-GFP. This data confirmed that the BC box domain and not the Ago-2 binding domain of CrPV-1A is required for SG inhibition.

## Mutations within CrPV-1A affect 2A peptide activity

To determine whether the effects on SG inhibition are due to differences in CrPV-1A protein levels, we monitored wild-type and mutant CrPV-1A protein levels in transfected cells by immunoblotting using anti-GFP and anti-CrPV-1A, which we raised against purified recombinant CrPV-1A protein (Fig 3A). For most mutant CrPV-1A, the individual CrPV-1A and GFP proteins were detected at similar levels after transfection, indicating that the wild-type and mutant CrPV-1A proteins are expressed and processed efficiently by the CrPV-2A 'stop-

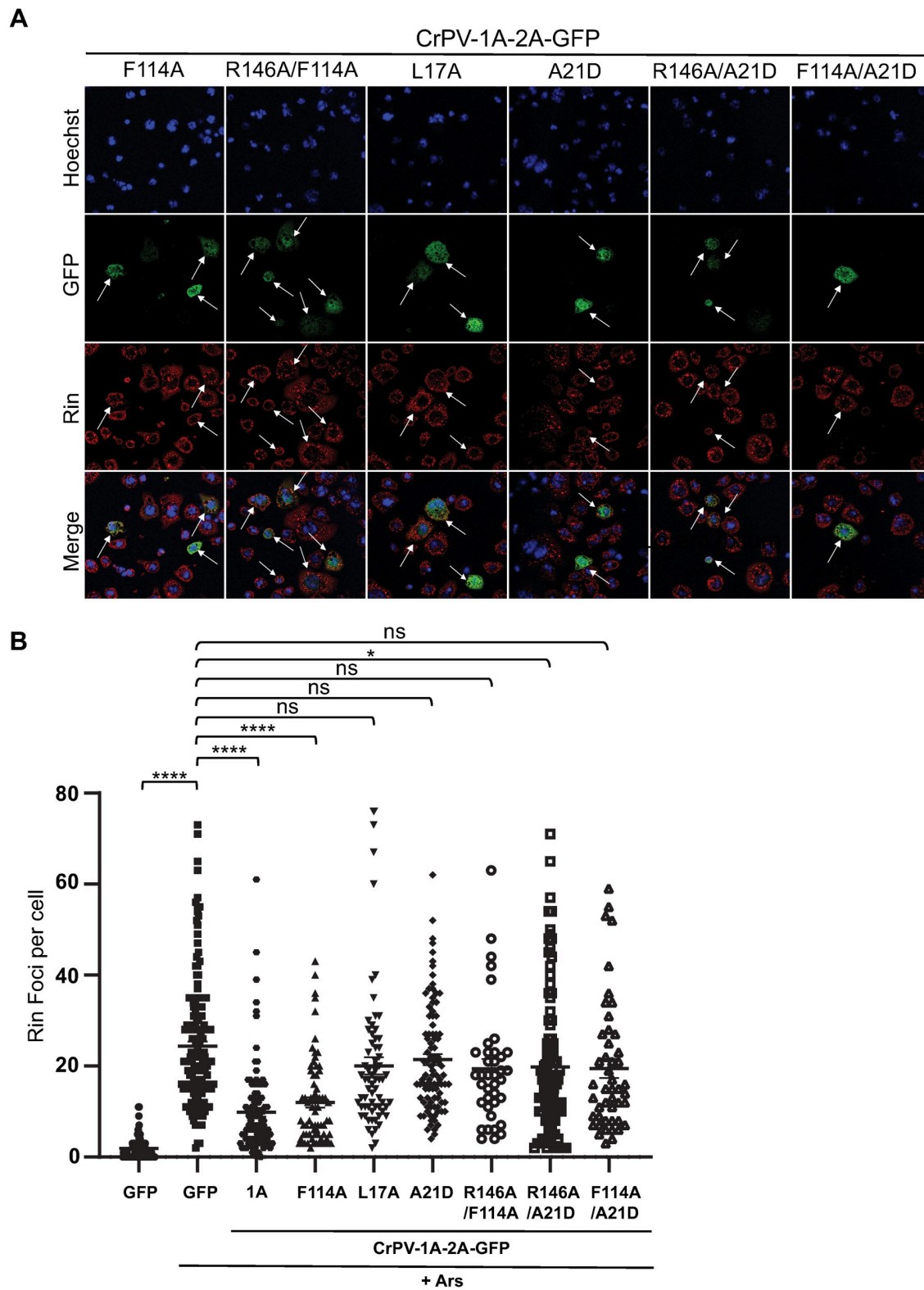

**Fig 2. CrPV-1A mediated stress granule inhibition requires the BC Box domain and is independent of the Ago-2 binding domain.** (A) Images of transiently transfected S2 cells with the indicated *in vitro* transcribed RNAs (16 hours), followed by one-hour sodium arsenite treatment (500 μM). Shown are representative transfected cells detecting GFP fluorescence (green), Rin antibody staining (red) and Hoechst dye staining for nucleus (blue) and merged images. The arrows show transfected cells. Images were taken

using the Leica Sp5 confocal microscope with a 63X objective lens and 2X zoom (B) Box plot of the number of Rin foci per cell. At least 50 cells were counted for each condition from three independent experiments. Data are mean ± SD. p > 0.05 (ns), p < 0.021 (*), p < 0.0001(****) by a one-way ANOVA (nonparametric) with a Bonferroni's post hoc-test.

go' translation activity (>99% efficiency). However, mutant L17A and A21D CrPV-1A were expressed at lower levels, suggesting that these mutations affected the stability of the proteins. In general, these results indicated that the effects of CrPV-1A on SG formation are not due to differences in protein levels. However, upon longer exposure, we observed that the mutant CrPV-1A(R146A) resulted in a slower migrating band with a mass that is predicted to be the unprocessed fusion CrPV-1A-2A-GFP protein (Fig 3A). To further confirm these results, we

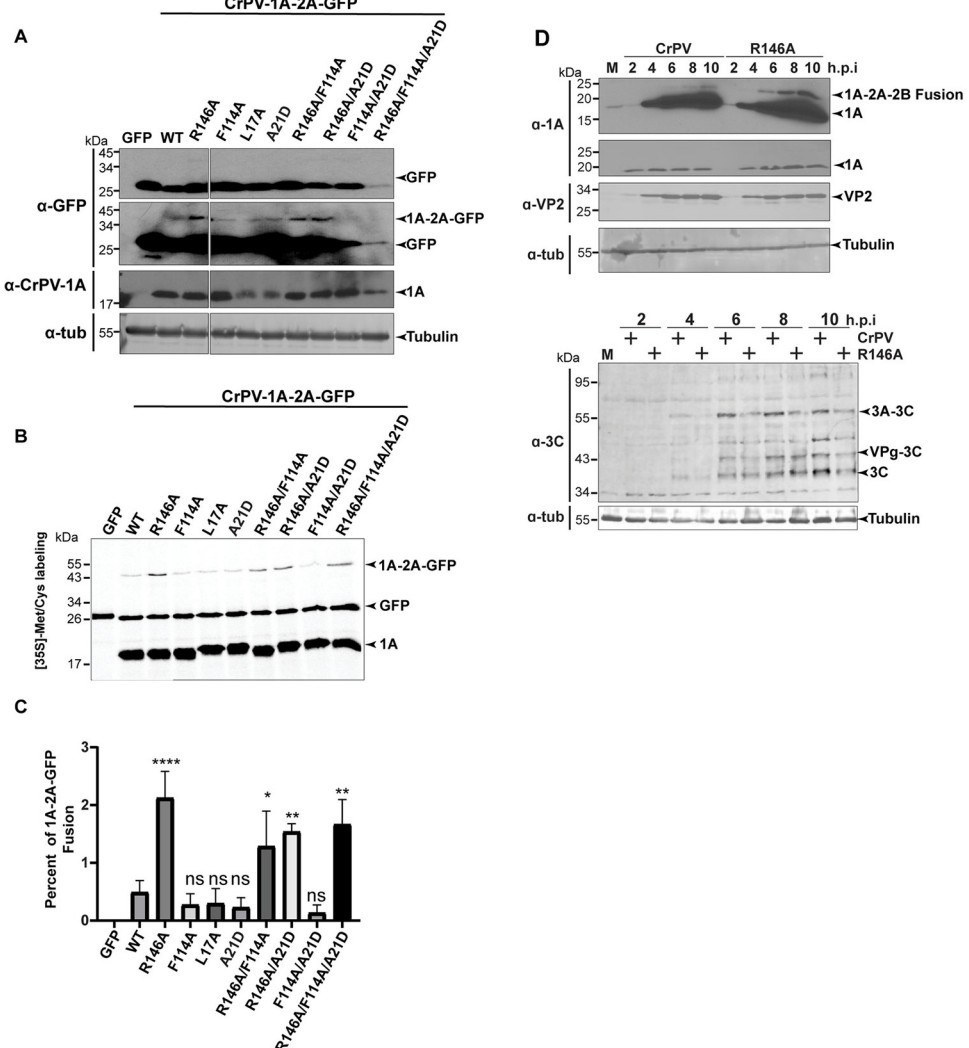

**Fig 3. R146 of CrPV-1A promotes full 2A peptide activity:** (A) Immunoblots of lysates from S2 cells transfected with the indicated reporter RNAs (16 hours post transfection). A light exposure (top) and longer exposure (bottom) of an anti-GFP immunoblot and CrPV-1A immunoblot is shown. (B) Autoradiography of [35S] Met/Cys labelled proteins from *in vitro* sf-21 translation reactions incubated with the indicated RNAs. (C) Percent quantification of [35S] Met/Cys labelled CrPV-1A proteins. Data are mean ± SD from three independent experiments. p > 0.05 (ns), p < 0.002 (**), p < 0.0001(****) by a one-way ANOVA (nonparametric) with a Bonferroni's post hoc-test. (D) Immunoblots of lysates from S2 cells infected with (M) mock, CrPV or CrPV (R146A) (MOI 10) at indicated time points.

directly monitored *in vitro* protein synthesis of the CrPV-1A-2A-GFP RNA in sf-21 insect lysates containing [$^{35}$S]-Met/Cys (Fig 3B and 3C). Similar to that observed in transfected cells, the majority of CrPV-1A mutants resulted in expression of separate CrPV-1A and GFP proteins, however, a minor unprocessed CrPV-1A-2A-GFP protein, ~2% of the total protein, was detected with the R146A mutation (Fig 3B and 3C).

To investigate this further, we monitored CrPV-1A expression in CrPV-infected S2 cells (MOI 10) using antibodies against non-structural proteins, CrPV-1A and the 3C-like protease and the structural protein, VP2 (Fig 3D). Immunoblotting showed that VP2 expression is reduced in CrPV(R146A)-infected cells, as shown previously [64]. The CrPV-3C-like protease antibody detected several unprocessed precursor polyproteins and the processed protein starting at 4 hours post infection (h.p.i) and increased over the course of infection. The 3C-like protein and its precursors were decreased in CrPV (R146A)-infected cells compared to that of wild-type infection (Fig 3D). The wild-type and mutant CrPV-1A(R146A) proteins were both expressed and processed to similar levels during infection. However, upon examining for the presence of a CrPV-1A-2A-2B fusion protein, we clearly detected a slower migrating band in CrPV(R146A)-infected cells. This result was similar to that observed in the overexpression and *in vitro* translation experiments (Fig 3B) and in support of the conclusion that the R146 residue is required for full CrPV 2A peptide activity.

## CrPV-1A expression leads to nuclear poly(A)+ RNA accumulation

Besides SG protein markers, poly(A)+ RNA is a key SG marker which contributes to the structural scaffold for SG assembly [2,10]. Although CrPV infection inhibits the assembly of Rin foci and other SG protein markers, cytoplasmic poly(A)+ RNA foci are still detected during both wild type and mutant CrPV (R146A) virus infection [64,69]. We investigated the effect of wild-type and mutant CrPV-1A expression on poly(A)+ RNA by monitoring poly(A)+ RNA localization using oligo-dT fluorescent *in situ* hybridization (FISH). In S2 cells transfected with control GFP RNA, the poly(A)+ signal was distributed in both the nucleus and cytoplasm (Fig 4A and 4B). Interestingly, S2 cells expressing wild-type CrPV-1A resulted in enrichment of poly(A)+ RNA signal in the nucleus. Quantification of the poly(A)+ signal in the nucleus compared to the total intensity in the cell showed that there is a reproducible difference in the distribution of the nuclear poly(A)+ signal in the control GFP expressing cells- vs the CrPV-1A-2A-GFP-transfected cells (Fig 4A and 4B). By contrast, poly(A)+ RNA was evenly distributed in both the nucleus and cytoplasm in S2 cells expressing mutant CrPV-1A(R146A)-2A-GFP. In cells expressing CrPV-1A(F114A)-2A-GFP, poly(A)+ RNA showed nuclear enrichment, similar to that observed when wild-type CrPV-1A is expressed (Fig 4A and 4B). These results suggested that expression of CrPV-1A modulates nuclear event(s) such as mRNA export or mRNA processing, leading to the accumulation of poly(A)+ signal in the nucleus.

## CrPV-1A is localized to the nuclear periphery in CrPV-infected S2 cells

Immunofluorescence analysis of CrPV-1A using anti-CrPV-1A showed that wild-type protein accumulates in the nucleus (Fig 4A), thus suggesting that the CrPV-1A nuclear enrichment may be linked to the poly(A)+ RNA signal in the nucleus. To investigate this further, we examined CrPV-1A localization in CrPV infected S2 cells by monitoring Z-stack immunofluorescence confocal images. In wild-type CrPV-infected cells, CrPV-1A was detected in both the nucleus and cytoplasm with a more enriched signal around the nuclear periphery (Fig 5A). Co-staining with anti-nuclear lamin showed overlap with CrPV-1A further supporting that CrPV-1A is located near the nuclear periphery (Fig 5A). Furthermore, CrPV-1A localization

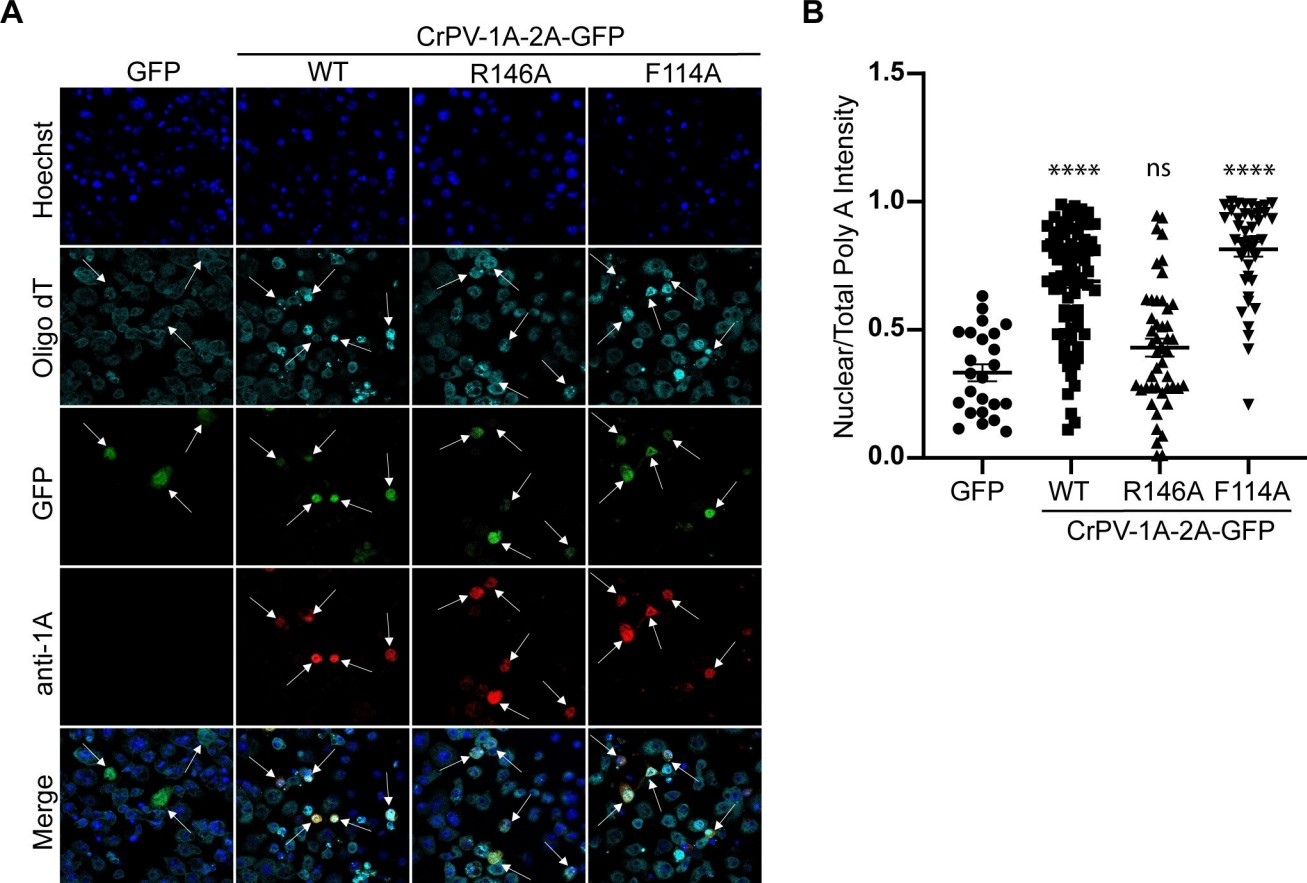

**Fig 4. CrPV-1A localizes to the nucleus and induces poly(A)+ RNA accumulation in the nucleus.** (A) Confocal immunofluorescence images of S2 cells transfected with *in vitro* transcribed RNA encoding CrPV-1A, CrPV-1A(R146A), or CrPV-1A(F114A) for 16 hours. GFP fluorescence (green), CrPV-1A antibody staining (red), fluorescence *in situ* hybridization using Cy5-oligo(dT) probes (cyan) and Hoechst dye (blue). The arrows show transfected cells. Images were taken using the Leica Sp5 confocal microscope with a 63X objective lens and 2X zoom (B) Box plot of the fraction of nuclear to total Cy5-oligo (dT) fluorescence intensity in each cell. At least 50 cells were counted for each condition from two independent experiments. Data are mean ± SD. p > 0.05 (ns), p < 0.021 (*), p < 0.002(**), p < 0.0001(****) by a one-way ANOVA (nonparametric) with a Bonferroni's post hoc-test.

to the nuclear membrane was observed at all time points during virus infection (S1 Fig). In CrPV(R146A)-infected cells, the CrPV-1A protein was distributed throughout the cell with limited overlap with nuclear lamin staining (Fig 5B). These results were in line with the conclusion that a fraction of CrPV-1A is localized to the nuclear periphery in infected cells.

## Transcriptome analysis in CrPV and CrPV (R146A) virus infected S2 cells

Our results suggest that CrPV-1A localizes to the nuclear periphery and induces poly(A) + RNA accumulation in the nucleus. Previous studies showed that despite global transcriptional shutoff, a subset of genes is transcribed in CrPV-infected S2 cells in a CrPV-1A(R146)-dependent manner [64]. To gain further insights, we performed transcriptome profiling by RNA-seq analysis of wild-type CrPV and CrPV(R146A)-infected S2 cells at 2 and 4 hours post infection (h.p.i). We removed all the CrPV RNA reads for downstream analysis. Principal component analyses (PCA) indicate substantial difference in genes induced during wild-type CrPV infection, whereas CrPV(R146A) infection induced minimal changes in the gene expression (Fig 6A). We ranked genes by their standard deviation across the samples and used the top 1000 genes in hierarchical clustering (distance by correlation and average linkage). In

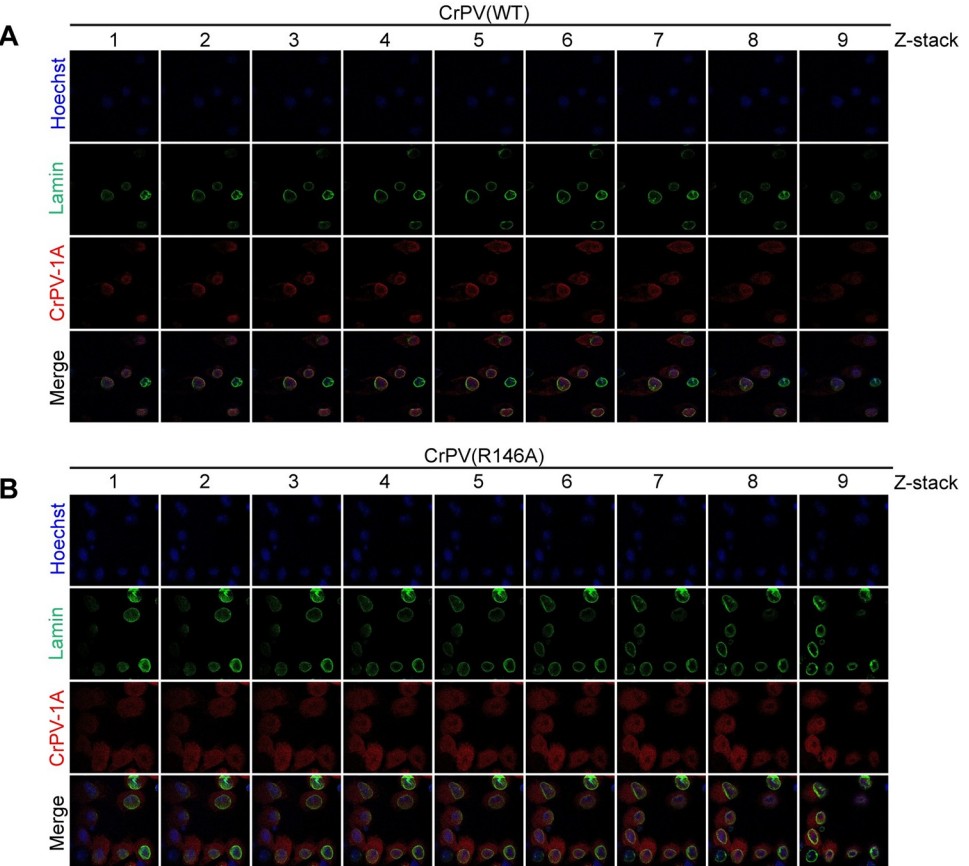

**Fig 5. CrPV-1A localizes to the nucleus during virus infection.** Z-stack confocal images of S2 cells infected with (A) wild-type CrPV or (B) CrPV(R146A) virus. From left to right are Z-stack images through the cells. Cells were fixed and stained with Lamin (green), CrPV-1A (red) and Hoechst (blue). Shown are representative images from three independent experiments.

agreement with the PCA analysis, our results suggest that wild-type CrPV, not the mutant CrPV(R146A)-infection induced substantial changes in steady state RNA levels (S2 Fig).

We observed a profound change on global steady state RNA levels in CrPV-infected cells at both 2 and 4 h.p.i. Specifically, at 2 h.p.i., wild-type CrPV infection resulted in dramatic changes in steady state RNA levels with 1325 genes showed increase and 1881 genes showed decrease in steady state RNA levels respectively, by two-fold compared to mock-infected cells. At 4 h.p.i., the effect became more prominent with ~2197 genes showed increase and ~2492 genes showed decrease in steady state RNA levels by two-fold. By contrast, CrPV(R146A) virus infection showed minimal changes in gene expression with only 25 genes and 175 genes resulting in a 2-fold increase and 18 and 78 genes showed 2-fold decrease in steady state RNA levels at 2 and 4 hours, respectively (Fig 6B–6E). Of those that were significantly altered under wild type virus infection at 4 hours (Fig 6B–6D), Gene ontology analysis filtered by molecular function revealed the upregulated genes are involved in cytoplasmic translation, peptide metabolic process, enzymatic activity and response to infection whereas the downregulated genes are involved in molecular functions such as RNA metabolic process and other macromolecular metabolic processes (Fig 6E).

We compared the transcriptome data with previous transcriptome data of S2 cells infected with DCV and flies infected with CrPV or DCV [70,71]. 38 out of 71 genes that were

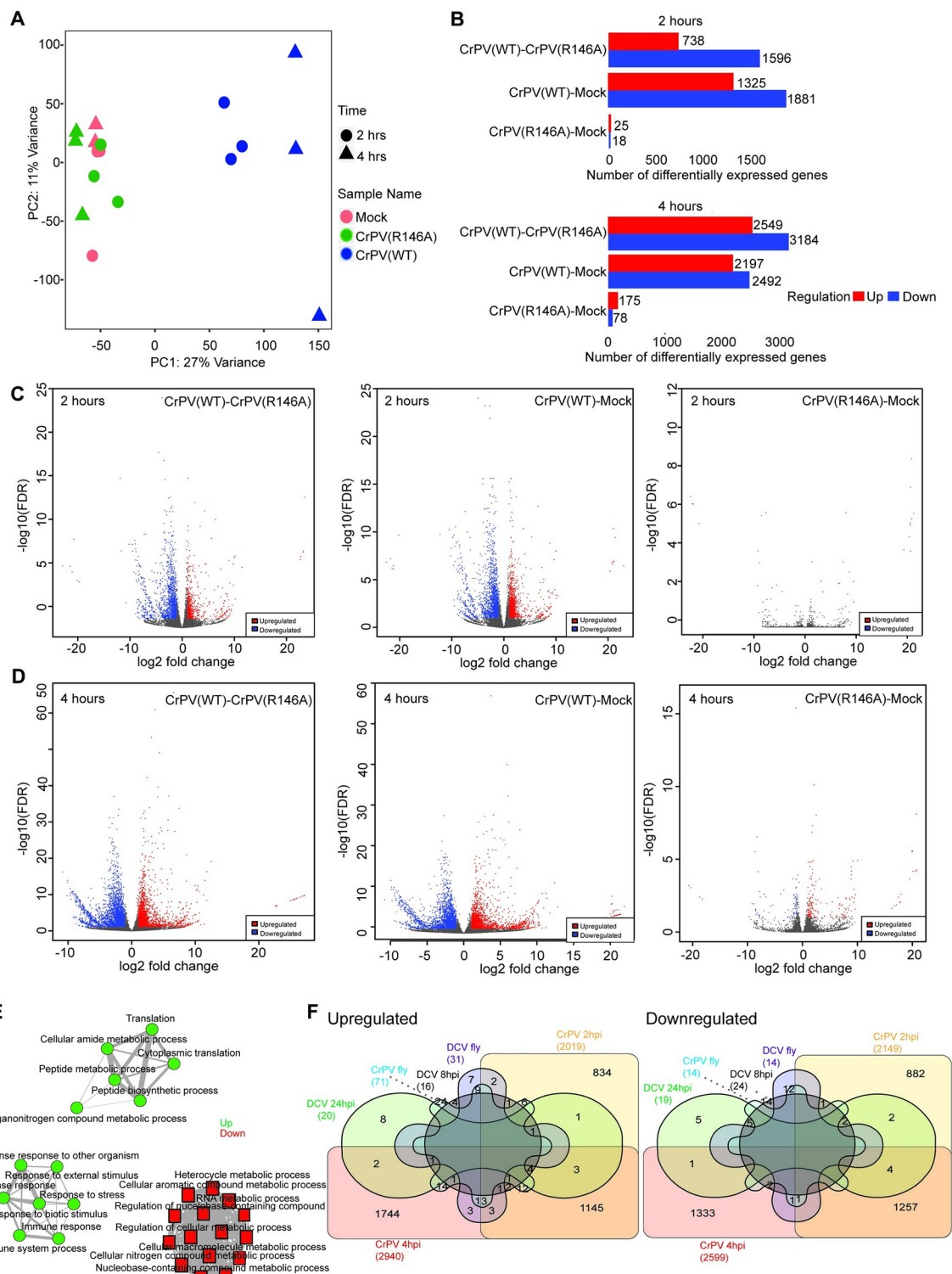

**Fig 6. Transcriptional profiling of CrPV and CrPV(R146A) infected S2 cells.** (A) Principal component analysis of transcriptional signatures from cells infected with Mock, CrPV or CrPV(R146A). (B) Bar graph indicating the number of differentially expressed genes for each comparison identified by DESeq2 (C) Volcano plots showing changes in gene expression with fold change (FC) in expression intensity of DEGs, plotted against corresponding FDR for 2 hrs and (D) 4 hrs. (E) Network analysis diagram showing Gene ontology analysis filtered by molecular function for CrPV infected cells (F) Venn diagram showing comparison of dicistrovirus transcriptome datasets [100].

upregulated in flies infected with CrPV showed upregulation and 2 out of 14 genes that were downregulated in flies infected with CrPV showed downregulation in our dataset. 12 out of 20 genes that are upregulated and 9 out of 19 genes that are downregulated in DCV infection at 24 hours are upregulated and downregulated respectively in CrPV infected S2 cells. There was no overlap between all the transcriptome datasets (Fig 6F). In summary, these results indicated that the CrPV-1A's R146 residue is essential in modulating the steady-state transcriptome in CrPV-infected S2 cells.

## Nucleo-cytoplasmic RNA export contributes to virus infection

Our results so far suggest that CrPV-1A modulates nuclear events to inhibit SG formation, possibly affecting mRNA export as there is an enrichment of poly(A)+ signal in the nucleus in CrPV-1A expressing cells. The NXF1-p15 heterodimer is a key mRNA export factor that promotes docking and transport of mRNA across the nuclear membrane via the nuclear pore complex to the cytoplasm [72]. We reasoned that if the mRNA export pathway contributes to CrPV infection, then depletion of NXF1 would affect CrPV infection. Incubation of S2 cells with NXF1 dsRNA but not control GFP dsRNA resulted in accumulation of nuclear poly(A)+ mRNA indicating that NXF1 knockdown was efficient in impairing mRNA export (Fig 7A and 7B) [73]. We then challenged control or NXF1 knockdown S2 cells with either wild-type CrPV or mutant CrPV (they R146A) virus infection (MOI 10). In control knockdown cells, CrPV(R146A) infection resulted in a decrease in viral titre compared to wild-type CrPV infection, as previously reported [64]. By contrast, in NXF1 KD cells, both wild-type and mutant CrPV (R146A) infection resulted in ~three-fold increase in viral titer (Fig 7C). These results suggested that mRNA export pathway contributes to CrPV infection.

## Nup358 is required for stress granule inhibition by CrPV-1A

A CrPV-1A interactome study identified host proteins involved in RNA export including mTor, GP210, Rae1, Nup88, Nup214 and Nup358 [59]. We reasoned that if the CrPV-1A (R146A) protein is defective in modulating the RNA export pathway, depletion of these factors would recover mutant CrPV(R146A) virus infection. To determine if these proteins contribute to CrPV infection, we depleted each mRNA by RNAi and then monitored wild-type or mutant CrPV(R146A) virus (MOI 1) infection by immunoblotting for viral VP2 protein expression. qRT-PCR analysis confirmed depletion of mRNAs by dsRNA treatment as compared to the control (dsFLuc) treated cells (S3 Fig). Knockdown of Rae1, mTor, Nup88 or Nup214 did not restore VP2 expression under mutant CrPV(R146A) virus infection compared to wild-type infection, similar to that observed in the control dsRNA treated cells (Fig 8A and 8B). By contrast, in GP210 or Nup358-depleted cells, VP2 expression was increased in mutant CrPV (R146A) infected cells to a level similar to wild-type CrPV infection, thus suggesting that replication was recovered (Fig 8A and 8B).

To support these findings, we performed qRT-PCR of RNA extracted from wild-type or mutant CrPV(R146A)-infected cells. As expected, CrPV RNA was reduced in CrPV R146A-infected cells compared to wild-type infection in control dsRNA treated cells. Depletion of GP210 resulted in decreased viral RNAs in CrPV(R146A)-infected cells, similar to the control, indicating a defect in replication (Fig 8C). However, in Nup358 knockdown cells, similar levels of CrPV RNA were recovered in wild-type and mutant CrPV(R146A)-infected cells (Fig 8C). These results strongly suggested that Nup358 is required for CrPV infection in an R146 residue-dependent manner.

To determine whether Nup358 promotes mRNA export and whether these effects are associated with SG formation, we monitored poly(A)+ RNA by FISH and arsenite- or pateamine

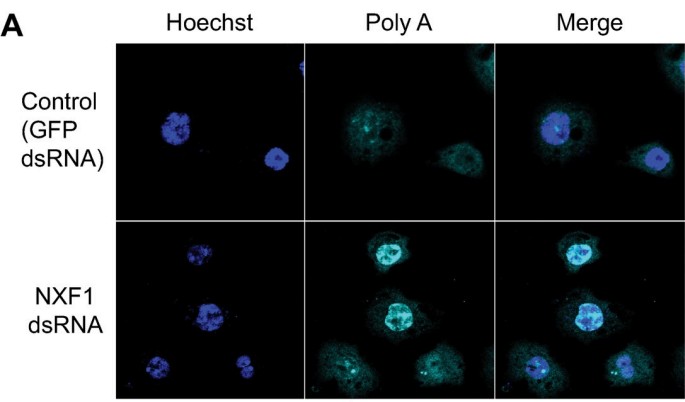

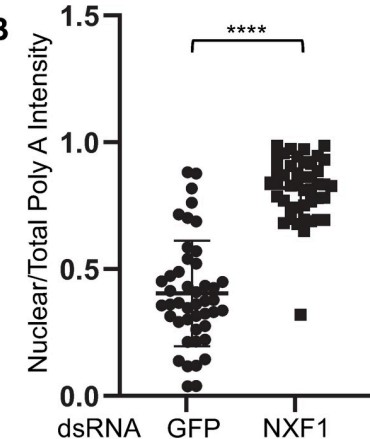

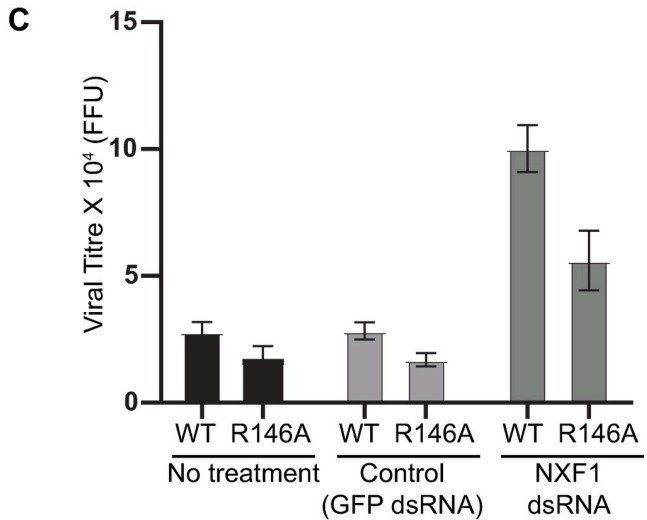

**Fig 7. RNA export modulates CrPV infection.** (A) Fluorescence *in situ* hybridization using Cy5-oligo(dT) (cyan) of S2 cells incubated with dsRNA targeting RNA export factor NXF1 or control GFP and Hoechst dye (blue). Images were taken using the Leica Sp5 confocal microscope with a 63X objective lens and 3X zoom. (B) Box plot of the fraction of nuclear to total Cy5-oligo(dT) fluorescence intensity in each cell. At least 40 cells were counted for each condition from two independent experiments. Data are mean ± SD. p < 0.0001(****) by a student t-test. (C) Viral yield from wild-type and mutant (R146A) CrPV infected S2 cells for 8 hours (MOI 10) was accessed by fluorescence foci unit (FFU). Shown are averages from two independent experiments.

A-induced Rin foci formation in Nup358 dsRNA-treated S2 cells. We generated an antibody raised against Drosophila Nup358. Immunoblotting for Nup358 showed a distinct protein band that migrated at >245 kDa, which was not detected in Nup358 dsRNA-treated cells, thus showing specificity of the antibody and confirming RNAi-mediated depletion of Nup358 (Fig 8D). Depletion of Nup358 resulted in poly(A)+ signal in the nucleus (Fig 8E and 8F), in line that Nup358 contributes to mRNA export in Drosophila cells [73]. Pateamine A treatment of Nup358-depleted cells resulted in decrease in number of Rin foci per cell (Fig 8E). Similarly, arsenite treatment of Nup358-depleted cells resulted in a reproducible decrease in the number of Rin foci per cell by ~40% as compared to control dsRNA treated cells (S4 Fig). Moreover, depletion of NXF1, the main RNA transport factor, resulted in loss of SGs, thus supporting the results that blocking mRNA export, in general, can lead to SG inhibition (S5 Fig). These results showed that Nup358 is necessary for pateamine A and arsenite stress-induced Rin foci formation in S2 cells and inhibition of RNA export contributes to SG formation.

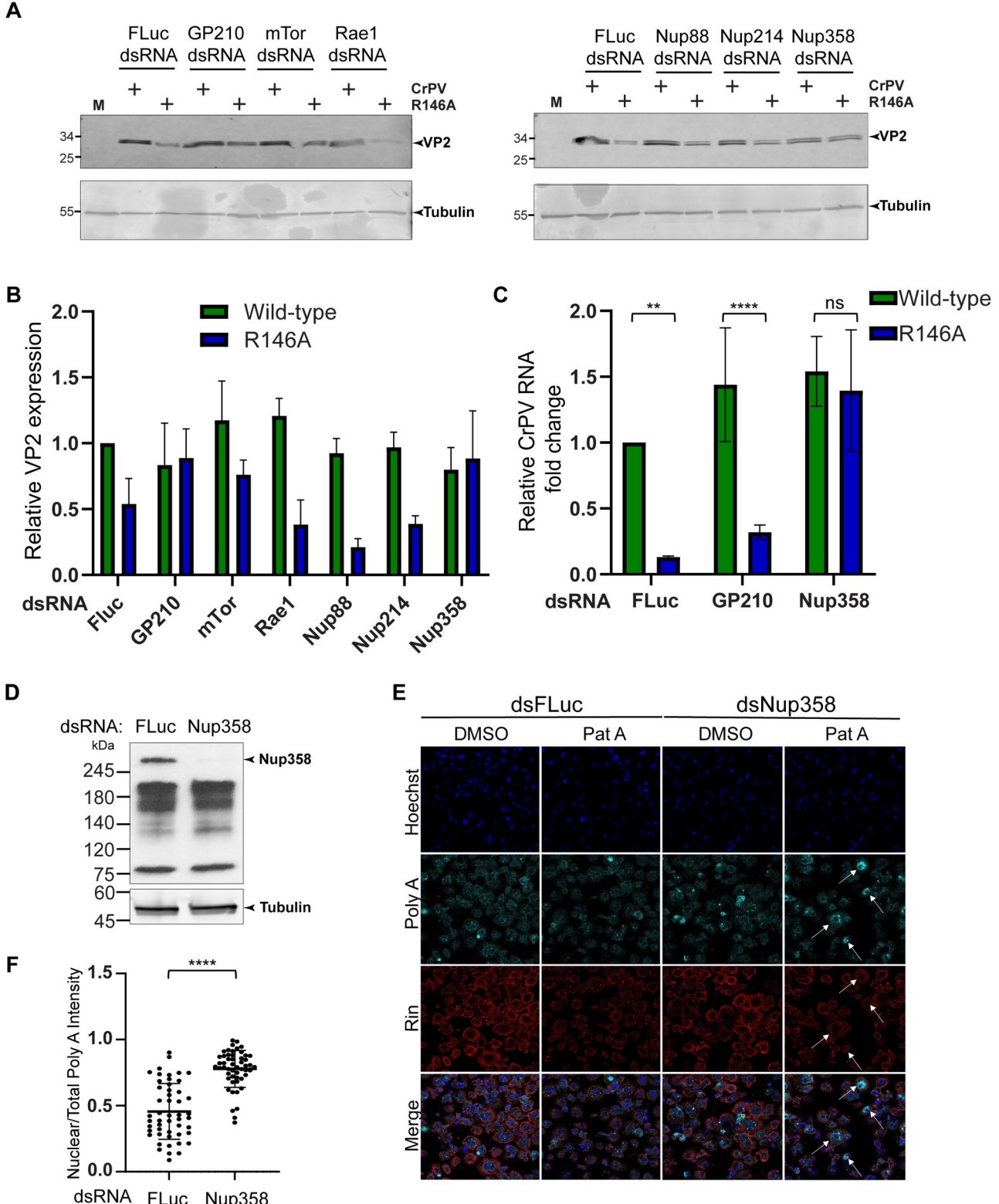

**Fig 8. Nup358 promotes CrPV infection in an R146-dependent manner.** (A) Immunoblots of S2 cells treated with dsRNA targeting GP210, mTor, Rae1, Nup88, Nup214, Nup358 or control FLuc, followed by mock infection or infection with wild-type or mutant CrPV virus for 8 hours (MOI 1). (B) Quantification of VP2 intensity normalized to tubulin from three independent experiments. The intensity values are normalized to the VP2/Tubulin intensity in FLuc control knockdown cells. (C) CrPV viral RNA levels by qRT-PCR analysis normalized to Rps9 mRNA levels. Data are mean ± SD

relative to WT p > 0.05 (ns), p < 0.002(**), p < 0.0001(****) by a one-way ANOVA (nonparametric) with a Bonferroni's post hoc test. (D) Immunoblot of S2 cell lysates treated with dsRNA targeting FLuc or Nup358. (E) FISH using Cy5-oligo(dT) or antibody staining of Rin (red) of S2 cells treated with control dsRNA or Nup358 dsRNA in the presence of DMSO or Pateamine A (Pat A). Hoechst staining is shown in blue. The arrows show Nup358-knockdown cells concomitant with nuclear poly(A)+ enrichment. Shown are representative images of at least two independent experiments. (F) Box plot of the fraction of nuclear to total Cy5-oligo(dT) fluorescence intensity in each cell. At least 50 cells were counted for each condition from two independent experiments. Data are mean ± SD. p < 0.0001(****) by a student t-test.

## CrPV-1A interacts with and targets Nup358 for proteasome-mediated degradation

Our results suggest that the wild-type, but not the R146A mutant CrPV-1A manipulates Nup358 to modulate SG formation. We next asked if CrPV-1A interacts with Nup358 and whether this interaction requires the R146 residue. To address this, we generated a *Drosophila* expression construct containing a 3XFLAG tagged-Nup358 (FLAG-Nup358). Transfection of S2 cells for 24 or 48 hours with this construct resulted in immunodetection of Nup358 with either FLAG antibody or Nup358 antibody, thus validating the stable expression of the full-length protein (S6 Fig). To address whether CrPV-1A interacts with Nup358, we transfected FLAG-Nup358 in *Drosophila* S2 cells for 24 hours followed by transfection with *in vitro* transcribed wild-type or R146A mutant CrPV-1A-2A-GFP RNA or 5'cap-GFP-poly(A) RNA for 16 hours. Expression of wild-type and mutant R146A CrPV-1A were similar. Immunoprecipitation using anti-FLAG to pulldown FLAG-Nup358 as bait resulted in immunoprecipitation of wild-type CrPV-1A (Fig 9A). By contrast, FLAG-Nup358 immunoprecipitates contained in less mutant CrPV-1A(R146A) protein, indicating that CrPV-1A interacts with Nup358 in an R146-dependent manner (Fig 9A).

Given that the CrPV-1A BC box domain is required for SG inhibition, we next investigated whether CrPV-1A mediates Nup358 degradation. In CrPV-infected cells, Nup358 protein levels were decreased as compared to mock-infected cells (Fig 9B). Notably, in mutant CrPV (R146A)-infected S2 cells, Nup358 protein levels were similar to mock-infected cells,

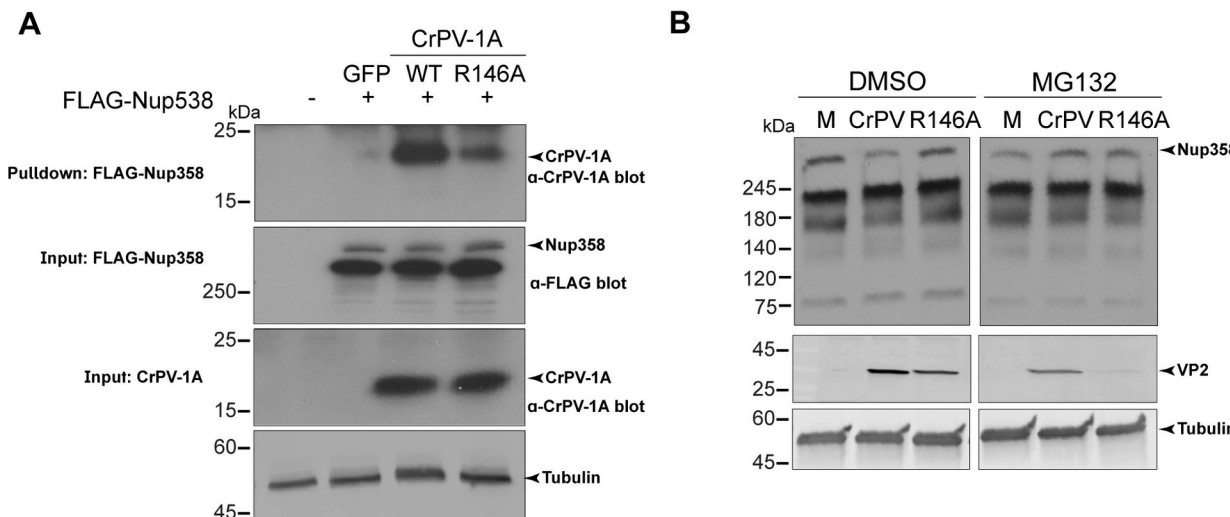

**Fig 9. Nup358 is degraded during CrPV infection.** (A) Co-immunoprecipitation assay of S2 cells transiently transfected with FLAG-Nup358 for 24 hours, then transfected with either GFP, CrPV-1A, or CrPV-1A(R146A) RNA for 16 hours. Lysates were incubated with FLAG antibody as bait and precipitated using Protein A/G Magnetic beads. Shown are representative immunoblots from three independent experiments. (B) Immunoblots of S2 cell lysates infected with (M) mock, CrPV or CrPV(R146A) (MOI 1) in the presence or absence of DMSO or 50 μM MG132. Shown are representative immunoblots from three independent experiments.

indicating that R146 of CrPV-1A is required for decreased steady-state levels of Nup358. To determine if Nup358 is degraded by the proteasome in CrPV-infected cells, we incubated CrPV-infected cells with the pan-proteosome inhibitor, MG132 [74]. Incubating MG132 in CrPV-infected cells led to slightly decreased VP2 levels as compared to DMSO-treated infected cells, indicating a minor inhibitory effect on CrPV infection (Fig 9B). By contrast, MG132-treated CrPV (R146A)-infected cells significantly inhibited VP2 expression, indicating sensitivity to proteosome inhibition compared to wild-type infection (Fig 9B). Importantly, treating CrPV-infected cells with MG132 recovered Nup358 protein levels to that of mock-infected cells, thus demonstrating that Nup358 degradation in CrPV-infected cells is proteo-some-dependent and supports the idea that CrPV-1A directly mediates Nup358 degradation to modulate SG assembly.

## Discussion

Inhibition of SGs is a general strategy employed by many viruses to facilitate infection [21]. The mechanism and consequences of SG inhibition during virus infection are not fully under-stood. In this study, we uncoupled the functions of the multifunctional CrPV-1A protein and reveal specific domains important for SG inhibition and virus infection. Specifically, we dem-onstrated that SG inhibition is dependent on the BC box domain of CrPV-1A, which recruits the Cul2-Rbx1-EloBC complex, and acts in concert with an essential R146 residue to promote infection. We provided insights into this mechanism by showing that the wild-type CrPV-1A but not mutant CrPV-1A(R146A) protein, localizes to the nuclear periphery, induces nuclear poly(A)+ RNA accumulation, and modulates global transcriptome changes. Finally, we showed that Nup358 is targeted for degradation by CrPV-1A in a R146-dependent manner. Together, we propose a novel viral strategy whereby the viral protein CrPV-1A targets Nup358 for degradation via its R146-containing C-terminal tail and recruitment of the Cul2-Rbx1-E-loBC complex inhibiting SG formation and RNA transport, consequently leading to poly (A) + mRNA in the nucleus that further contributes to SG inhibition and facilitate productive virus infection.

The effects of the R146A mutation on CrPV-1A's function are illuminating that point to a nuclear event(s) controlled by CrPV-1A that are likely interdependent. Besides disrupting CrPV-1A's ability to block SG assembly and Ago-2 activity, the CrPV-1A protein localizes to the nuclear periphery and mediates poly (A)+ mRNA nuclear enrichment and global tran-scriptome changes under infection (Figs 4–6). As mRNAs act as scaffolds for SG assembly [18,64,69,75], the enrichment of poly (A)+ mRNA in the nucleus in CrPV-1A expressing cells may serve two purposes: 1) to block global host mRNA translation and antiviral responses and 2) to deplete mRNA from the cytoplasm leading to SG inhibition. This viral strategy is remi-niscent of other viral proteins that modulate nuclear events to facilitate SG formation and infection. As examples, picornavirus 2A protease expression regulates SG assembly and RNA export [76,77] and influenza virus polymerase-acidic protein-X (PA-X) protein inhibits SG formation concomitant with cytoplasmic depletion of poly(A) RNA and accumulation of poly (A) binging protein (PABP) in the nucleus [78]. There is also precedent that modulation of mRNA export can affect SG formation. A recent study showed that blocking mRNA export pathways with Tubercidin, an adenosine analog, induces SG formation, likely indirectly regu-lating cytoplasmic protein synthesis [79]. Conversely, sequestering mRNA export factors into SGs can inhibit nucleocytoplasmic transport [80]. In this study, we present a new paradigm of SG inhibition by a viral protein that directly modulates nuclear mRNA export.

How does CrPV-1A regulate multiple cellular processes? Even though CrPV-1A is only 166 amino acids in length, there are multiple domains that mediate specific cellular functions. One

of the best-known functions of CrPV-1A is its ability to bind to Ago-2 via its TALOS domain and degrade Ago-2 by recruiting the Cul2-Rbx1-EloBC via its BC Box domain [59]. By systematically uncoupling the functions of CrPV-1A via specific mutations singly or in combination with R146A, we demonstrate that CrPV-1A's ability to block SG formation is not dependent on its ability to bind to Ago-2 (F114A mutation) (Fig 3). It is also clear that the TALOS domain does not contribute to CrPV-1A's effects on enrichment of nuclear poly(A)+ mRNA (Fig 4). However, our results point to a role of the BC Box domain as mutations within this domain resulted in a deficit in SG inhibition by CrPV-1A. These results strongly suggest that recruitment of Cul2-Rbx1-EloBC ubiquitin ligase complex is required for CrPV-1A's effects on SG inhibition. Further, our results identified Nup358 as a key component in inhibiting SG formation by CrPV-1A and promoting CrPV infection. Nup358 (also known as RanBP2) is an integral component of the cytoplasmic filaments of the nuclear pore complex that mediates nucleocytoplasmic transport of mRNA and protein [73,81]. Indeed, depletion of Nup358 in Drosophila cells blocks mRNA export from the nucleus (Fig 8) [73]. Recent studies have shown that Nup358 localizes to SGs [80,82]. Moreover, Nup358 plays a prominent role in

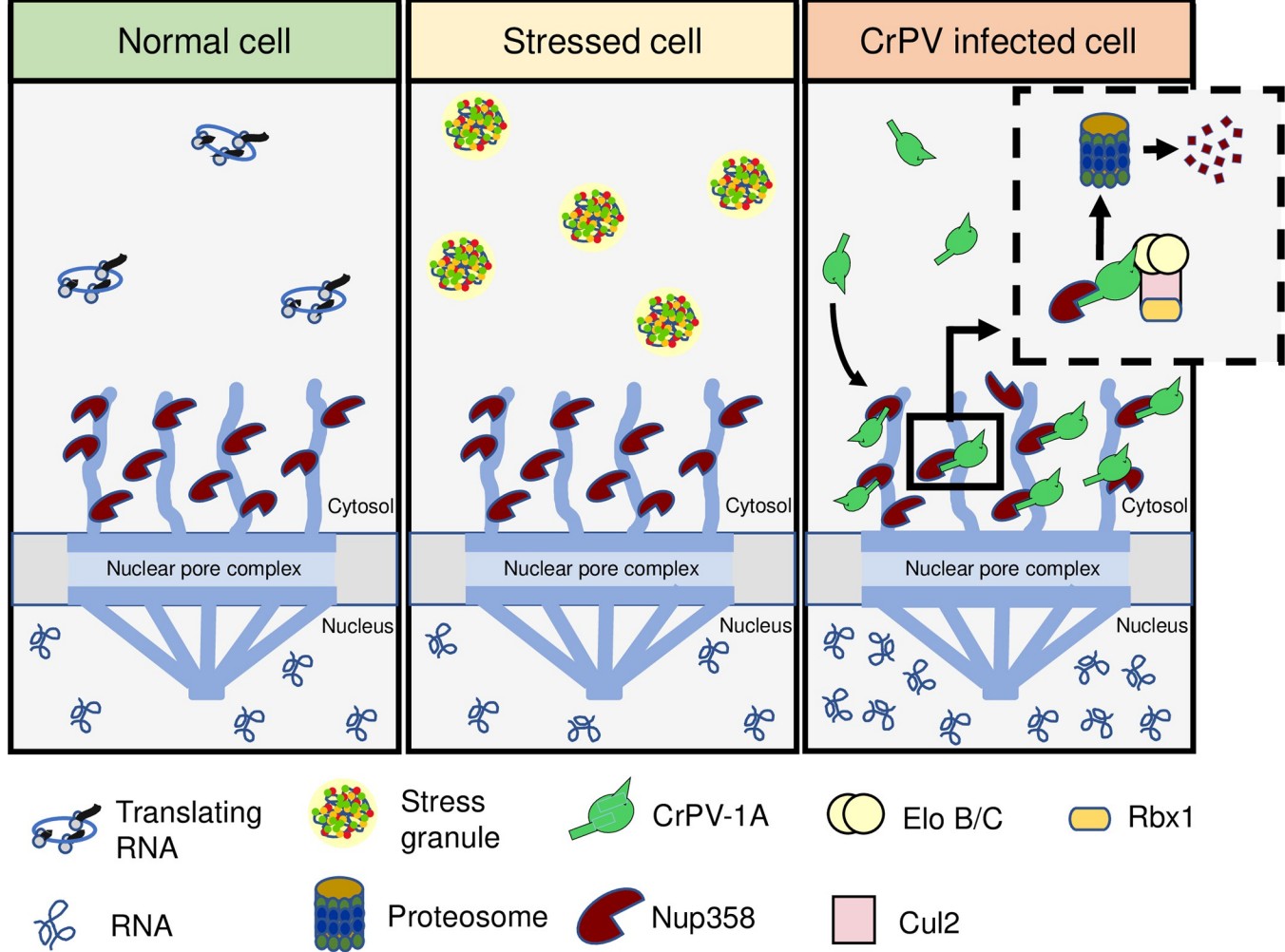

**Fig 10. Model of stress granule inhibition by CrPV-1A.** During CrPV infection, CrPV-1A localizes to the nuclear membrane in an R146-dependent manner and recruits Cul2-Rbx1-EloBC complex to ubiquitinate Nup358 leading to its degradation. The degradation of Nup358 results in a block in mRNA export, resulting in the enrichment of poly (A)+ mRNAs in the nucleus, and together inhibits stress granule formation which, facilitates virus infection.

virus infections. In HIV infected cells, Nup358 facilitates transport of the viral genome into the nucleus [83]. Additionally, vaccinia virus recruits Nup358 to the viral factories to enhance virus infection [84]. In this study, our model posits that CrPV-1A recruitment of the Cul2-Rb-x1-EloBC ubiquitin ligase complex targets Nup358 for proteosome-dependent degradation to inhibit mRNA export and subsequently, to block SG formation (Fig 10). In support of this model, Nup358 levels, which are decreased in CrPV-infected cells, are recovered in MG132-treated cells and depletion of Nup358 results in inhibition of stress-induced SG formation (Figs 7 and 8). We note that MG132 treatment decreases CrPV infection in general, which is similar to that observed with other plus strand RNA virus infections [85]. It is likely that proteasome activity is required for CrPV infection. For example, Ago-2 degradation by CrPV-1A requires proteasome activity [59]. Moreover, all of these effects are dependent on the R146 residue within the C-terminal tail of CrPV-1A. The R146A mutation may disrupt CrPV-1A/Nup358 interactions (Fig 9) or may alter CrPV-1A subcellular localization and thereby sequester it from interacting with Nup358. Finally, the effects of the R146A mutation may alter protein conformations that mediate these effects. Our model showed that the CrPV-1A C-terminal tail interacts directly or indirectly with Nup358 to mediate degradation by the Cul2-Rbx1-EloBC ubiquitin ligase complex. CrPV-1A acts as a hub that interacts with multiple partners and through recruitment of the Cul2-Rbx1-EloBC ubiquitin ligase complex leads to proteosome-dependent degradation [59]. It will be of interest to investigate in more detail of the CrPV-1A/Nup358 interactions and whether ubiquitination of Nup358 is required for degradation or inactivation. Although the mammalian Nup358 is also a small ubiquitin-like modifier (SUMO) E3 ligase, that can SUMOylate Ago-2 and is linked to SG dynamics [86–90], however the Drosophila Nup358 lacks an obvious sumoylation domain [73].

Besides affecting the above cellular processes, the CrPV-1A R146A mutation also modulates the CrPV-2A peptide stop-go activity (Fig 2). 2A peptide activity relies primarily on a conserved DxExNPGP sequence whereby the stop-go peptidyl-tRNA hydrolysis occurs between the last G and P [60]. However, sequences upstream of this conserved region also contributes to 2A activity [91,92]. The R146A mutation, which is 20 amino acids upstream of the "stop-go" cleavage, would still be within the ribosome exit tunnel during translation and thus, specific peptide-ribosome exit tunnel interactions likely affects CrPV-2A activity. Although the inhibitory effects on 2A activity is modest (~2%), it is possible that the expression of the fusion CrPV-1A-2A-2B protein may act in a dominant manner to the effects observed by CrPV-1A (R146A) expression including mRNA export, Nup358 degradation and SG formation, ideas that needs to be examined further.

The small CrPV-1A protein employs a multi-prong 'Swiss-army knife' approach to block the insect antiviral response, transcription, RNA metabolism and SG formation, all of which facilitate infection [64]. SGs are "sinks" of RNA and protein that may sequester viral proteins and RNA that may delay virus infection. For example, the CrPV 3C protease can localize to SGs [69]. SG inhibition is likely a key viral strategy to ensure viral proteins and RNA are available to promote the viral life cycle. Our previous study also showed that CrPV-1A can also block SGs in human cells, thus it will be interesting to determine whether there are common mechanisms for SG inhibition by CrPV-1A across species [64]. Although sequence analysis of the other dicistrovirus 1A proteins do not show any obvious conservation, it has been shown that some may have similar functions; the related DCV-1A protein can inhibit the antiviral RNAi pathway through a distinct mechanism by binding to dsRNA [58]. It will be of interest to determine whether these other dicistrovirus 1A proteins act similarly as CrPV-1A, which may shed light into the diversity of protein domains that target specific host factors for productive virus infection. Given the growing global health concerns of arthropod-borne viruses such as Zika virus, Dengue virus and Chikungunya virus, it will be of importance to further

understand the underlying fundamental virus-host interactions such as stress granule inhibition in insects in order to develop novel antiviral strategies.

## Materials and methods

### Cell culture and virus infection

Drosophila S2 cells (Invitrogen) derived from a primary culture of late-stage *Drosophila melanogaster* embryo were maintained and passaged in either Shields and Sang medium (Sigma) or Schneider's medium (Thermo Fisher Scientific) supplemented with 10% fetal bovine serum (Gibco) and 1X Penicillin-Streptomycin at 25˚C.

The wild-type and mutant CrPV clones [93] were used to prepare virus stocks and the stock was expanded by reinfecting naïve cells as described previously [56]. S2 cells were infected with wild type or mutant virus at the desired multiplicity of infection in phosphate buffer saline (PBS) at 25˚C. After 30 mins of absorption, complete medium was added, and cells were harvested at desired time points. Virus titers were determined by Fluorescence Foci Forming Unit assay using immunofluorescence (anti-VP2) as previously described [56].

### Plasmids

The Drosophila expression vector pAc 5.1/V5-His B containing CrPV 5'UTR-GFP-3' UTR, CrPV 5'UTR-1A-GFP-3' UTR was generated using Gibson assembly (NEB Gibson assembly). The respective mutants were generated using Site directed mutagenesis. dsRNA targets were selected chosen from Updated Targets of RNAi Reagents Fly (Flybase) Fragments of candidate genes NXF1 (Accession number: AJ318090.1;position:2056–2362), GP210 (Accession number: AF322889.1; position: 2077–2576), mTor (Accession number: NM_057719.4; position:462–961), Rae1 (Accession number: NM_CP023332.1; position:17547164–17546605), Nup88 (Accession number:AY004880.1; Position: 116–714), Nup214 (Accession number: NM_143782.3; Position: 1412–1853) and Nup358 (Accession number: NM_143104.3; position: 1768–2338) targeting all isoforms with no off targets were used for dsRNA mediated knockdown. The amplicons were synthesized as gene fragments (Twist Bioscience) containing a T7 polymerase promoter flanking either side of the amplicon and directly cloned into a pTOPO plasmid using EcoR1 (NEB). FLuc plasmid was described previously [59,64]. The plasmids were digested with EcoRI, and the digested and purified products were directly used for *in vitro* transcription reactions. Gene encoding the Drosophila Nup358 transcript variant B (Accession number NM_001260373.2) with 3X-N terminal FLAG tag was cloned between BamHI and XhoI sites in pAc/His B vector (Genscript). All plasmids were sequence confirmed by Sanger sequencing (Genewiz).

### *In vitro* transcription and translation

T7 polymerase reactions were performed as described previously [94]. Briefly 5 µg pAc CrPV 5'UTR-1A-GFP-3' or pCrPV-3 plasmids were linearized with Eco53KI (NEB) or 5 µg pTOPO dsRNA plasmids with EcoR1 (NEB) in reaction containing 1X T7 buffer (50mM Tris-HCl, 15mM $MgCl_2$, 2 µM Spermidine Trihydrochloride, 5 µM DTT), 10 mm NTP mix (NEB), Ribolock (Thermo scientific), 2 units of yeast inorganic pyrophosphate (NEB) and T7 Polymerase for 4–6 hours. DNAse I (NEB) treated samples were cleaned up using RNAeasy cleanup kit (Qiagen). GFP RNA was capped and polyadenylated (Cellscript) and then purified (RNAeasy kit, Qiagen). The integrity of RNA was verified by denaturing RNA agarose gel electrophoresis. The quantity of RNA was determined using Nanodrop (Thermo Scientific).

*In vitro* translation assays of the wild type or mutant CrPV-1A RNAs were performed in *Spodoptera frugiperda* 21 (sf-21) insect cell extract (Promega). Briefly, 2 µg RNA was incubated

with sf-21 extract in the presence of $[^{35}S]$-Methinone-Cysteine (Perkin-Elmer >1000Ci/ mmol) and F buffer (40mM KOAc. 0.5 mM $MgCl_2$) for 2 hrs at 30°C. The resulting translated proteins were resolved by a sodium dodecyl sulfate (SDS)- polyacrylamide gel electrophoresis (PAGE) gel and analyzed by phosphoimager analysis (Typhoon, Amersham, GE life sciences).

## Transfections

For DNA transfections, 1.5 million S2 cells were transfected with 2 μg of plasmid using Xtreme-GENE HP DNA transfection reagent (Roche) according to the manufacturer's protocol. Transfected cells were incubated in complete Shields and Sang medium for 16–24 hours. Transfection of *in vitro* transcribed RNAs in S2 cells performed using Lipofectamine 2000 (Invitrogen) as described by manufacturers protocol for 16–24 hours.

## RNA interference

For dsRNA mediated gene knockdown, 3 million cells were incubated with serum free medium containing 60 μg dsRNA per well of a 6 well plate for 1 hour at 25°C. The soaked cells were supplemented with complete medium containing FBS and incubated for 4 days at 25°C. Cell viability of silenced cells were monitored by Trypan Blue dye exclusion assay.

## Immunofluorescence and *in situ* hybridization

Transfected S2 cells transferred to coverslips precoated with 0.5 mg/mL Concanavalin A (Calbiochem) in 12 well plates for 2 hours. 16–24 hours post transfection the cells were fixed in 3% w/v paraformaldehyde in PBS, then permeabilized in PBS containing 0.1% Triton X-100 for 30 minutes and blocked with 2% BSA for 30 minutes.

For *in situ* hybridization, the cells were incubated in hybridization buffer (2X SSC, 20% formamide, 0.2% BSA, 1 μg/μL yeast tRNA) for 15 mins at 37°C and subsequently, the cells were incubated with 1 mg/mL oligo(dT) conjugated to Cy5 (IDT) overnight at 46°C in hybridization buffer. The next day, the cells were washed with 2X SSC with 20% formamide twice for 5 min each at 37°C, 2X SSC for 5 min at 37°C, 1X SSC once for 5 min and 1X PBS for 5 min prior to staining with the primary antibodies.

The primary antibodies and the dilutions used were as follows: α-CrPV-1A (1:200), α-Lamin A (1:1000, DGRC), α-Rin (1:500, generous gift from Eric Lecuyer). Cells were washed three times with PBS and then incubated with secondary antibody (1:1000 goat anti-rabbit antibody or goat anti-mouse antibody conjugated to Texas Red and 1:1000 goat anti-mouse antibody conjugated to Alexa Fluor 647 (Life Technologies) and 1:1000 donkey anti-goat antibody conjugate to Texas Red (Thermo Fisher Scientific) and Hoechst dye (1:20,000 in PBS, Invitrogen) to stain for nuclei. Coverslips were mounted on slides with Prolong gold antifade reagent (Invitrogen). The cells were imaged and analyzed using a Leica SP5 confocal microscope (Leica Microsystems, Wetzlar, Germany) with a 63x objective. Representative images are shown and were analyzed in ImageJ.

Rin granules were counted using a quantitatively measured threshold intensity and defined circularity using Image J. Intensity measurements were done using Image J [95]. Box plots and graphs generated using GraphPad Prism is used to represent the data.

## cDNA synthesis and quantitative real time PCR

Total RNA was extracted from cells using Monarch total RNA Miniprep kit (NEB). cDNA synthesis was performed using Lunascript RT Supermix Kit (NEB) as per manufacturer's protocol. qRT PCR was performed using Luna Universal qPCR master mix (NEB) as per

manufacturer's protocol. CrPV genome was amplified using; 5'-CAGTGCCTTACATTGCC A-3' and 5'-AACTTCTACTCGCACTATTC-3' and Rps6 was amplified using primers 5'-CGA TATCCTCGGTGACGAGT-3' and 5'-CCCTTCTTCAAGACGACCAG-3'

## Western Blot analysis

S2 cells were washed with PBS and harvested in RIPA buffer (150 mM NaCl, 1% IGEPAL CA-630, 0.5% sodium deoxycholate, 0.1% SDS, 10% glycerol, 50 mM Tris-HCl, pH 8.0 and protease inhibitor cocktail (Roche)). Protein samples that were freeze thawed three times,spun down at 13,000 rpm for 15 minutes at 4°C and the supernatants were collected as the total protein extracts. Protein concentration was determined by Bradford assay (Biorad). Equal amounts (in micrograms) of lysates were separated on 4–15% SDS-PAGE gel and transferred to polyvinylidene difluoride (PVDF) membrane (Millipore). Subsequently, the membranes were blocked for 30 mins in 5% skim milk and TBS-T (20mM Tris, 150mM NaCl, 0.1% Tween-20) and probed with primary antibody for 1 hour.

The dilutions and primary antibodies used were as follows: α-CrPV-1A (1:1000), α-GFP (1:1000, Roche). α-CrPV-VP2 (1:1000, Genscript), α-CrPV-3C (1:1000, (raised against CrPV-3C peptide sequence NH$_2$-CTDMFDYESESYTQR-C), Genscript), α-CrPV-Nup358 (1:1000. raised against Nup358 peptide sequence NH$_2$-CGSTDKSEPGKDAGP-C), Genscript), α-Tubulin (1:1000, DSHB). Membranes were washed with TBS-T three times and incubated with secondary antibodies for 1 hour at room temperature. Following secondary antibodies were used: IRDye 800CW goat-anti-rabbit IgG or IRDye 680CW goat anti-mouse at 1:5000 (LI-COR Biosciences). Membranes were washed with TBS-T three times and protein bands were detected and quantified using the Odyssey Infrared Imaging System (LI-COR Biosciences). Alternatively, 1:5,000 dilution of donkey anti-rabbit IgG-horseradish peroxidase (Amersham) or a 1:5,000 dilution of goat anti-mouse IgG-horseradish peroxidase (Santa Cruz Biotechnology) was used to detect proteins by enhanced chemiluminescence (Thermo Scientific).

## Co-immunoprecipitation

Drosophila S2 cells were transfected with 3X-FLAG-Nup358 DNA for 24 hours, followed by transfection with in vitro transcribed RNA encoding GFP, CrPV-1A or CrPV-1A(R146A) for 16 hours. Cells were washed with PBS, lysed in the Pierce-MS Compatible Magnetic IP Kit lysis buffer, and freeze thawed three times to extract the total protein. After centrifugation, the supernatant was collected, and protein concentrations were determined by Bradford assay (Biorad). Co-immunoprecipitation was performed using Pierce-MS Compatible Magnetic IP Kit (Thermofisher) according to the manufacturer's protocol with minor modifications. Briefly 25 μL magnetic beads were incubated with 5 μg mouse monoclonal FLAG antibody (Sigma) at 4°C overnight. The beads were washed 2X times with an IP Lysis buffer and incubated with 1 mg total protein for 2 hours at 4°C overnight. Unbound proteins were washed off using wash buffers and the beads complexed with immunoprecipitated proteins were resuspended in the Pierce-MS Compatible Magnetic IP Kit elution buffer and processed for western blotting analysis.

## RNA seq: sample preparation, library generation and analysis

S2 cells (1.5 X 10$^7$) were infected with wild-type or R146A mutant CrPV at an MOI 3. Total RNA was extracted from mock or virus infected cells using Trizol reagent (Invitrogen) at 2 and 4 hours post infection. The samples were treated with DNAse I for 1 hour at 37°C and were re-extracted using Trizol. The RNA integrity was verified by denaturing gel analysis.

Polyadenylated RNA was isolated using NEBNext poly(A) mRNA isolation module and the quality and quantity of RNA were determined by electrophoresis on the bioanalyzer (Agilent). NEBNext Ultra II DNA library prep kit was used to generate libraries. Size selection was performed on adaptor ligated libraries using agarose gel, generating cDNA libraries size ranging from 150–275 nucleotides. The enriched libraries were purified using QlAquick purification column.

Sequencing of a pool of multiplexes libraries were performed on an Illumina HiSeq 4000 PE100 Platform (Génome Québec). At least 19 million reads were generated from each sample. Libraries, sequencing, and quality control of the sequencing were performed by the Nanq facility at Génome Québec.

Reads were trimmed based on quality using the default parameters of Trimmomatic and assessed using FastQC as part of Unipro UGENE v1.29 [96,97]. and mapped to the Drosophila melanogaster genome using default paired-end parameters of Bowtie2 as part of UGENE. Reads mapping to the CrPV genome were removed for downstream analysis to maintain normalization based only on total host gene transcript numbers. Reads were mapped to the *D. melanogaster* transcriptome and quantified using the quasi-mapper Salmon 1.8.0 [98]. Differentially expressed genes were identified using iDEP 0.92 (http://bioinformatics.sdstate.edu/idep92/) and the Bioconductor package DESeq2. used for heatmap visualization with Integrated differential expression and pathway analysis (iDEP) [99].

The raw sequencing data was submitted under Gene expression accession number PRJNA771107. Venn diagrams for the comparison of different gene expression data were generated using InteractiveVenn [100]. Network analysis of gene ontologies was performed using ClueGo v2.5.6 [101]as part of Cytoscape v3.7.2 [102] using the EBI-UniPRot-GOA Molecular Function database (17.02.2020).

## Anti-CrPV-1A Polyclonal antibody

DNA fragment encoding the full length CrPV-1A gene was cloned into pet28b vector using Nd1 and Xho1 enzymes and the resultant construct with C terminal His-tag was used for protein expression in *E.coli* BL21DE3 cells (modified from [63]). Expression of CrPV-1A protein was carried out in *E coli* (BL21DE3) cells grown in Terrific broth medium at 16˚C overnight. induced with 0.5 mM Isopropyl-β-D-thiogalactoside (IPTG). The soluble protein was purified using Ni-NTA Agarose beads (Qiagen) in a buffer containing 30 mM HEPES-KOH pH 7.4, 100 mM KOAc, 2 mM Mg(Ac)$_2$, 300 mM Imidazole, 10% glycerol, 1 mM DTT with complete mini EDTA free protease inhibitor tablet. The purified samples were dialyzed and further analyzed over a superdex 50 gel filtration column equilibrated with exchange buffer (30 mM HEPES-KOH pH 7.4, 100 mM KOAc, 2 mM Mg(OAc)$_2$, 10% Glycerol, 1 mM DTT). All purified proteins were flash-frozen in liquid nitrogen and stored at -80˚C. The polyclonal antibody against CrPV-1A in rabbits was generated by Genscript. USA.

## Supporting information

**S1 Fig. CrPV-1A localizes to the nucleus at times post-infection.** (A) Fluorescent images of S2 cells infected with CrPV (MOI 10) at indicated time points stained with CrPV-1A antibody (red) or Hoechst dye (blue).
(TIF)

**S2 Fig.** CrPV infection induces changes in gene expression (A) Hierarchical clustering of top 1000 genes (ranked by standard deviation across all the samples) showing difference in gene expression induced during virus infection. (B) Bar diagram showing comparisons on number

of differentially expressed genes (C) Network analysis on upregulated genes.
(TIF)

**S3 Fig. Knockdown of RNA export factors.** Bar graph showing relative mRNA levels by qRT-PCR in S2 cells treated with indicated dsRNAs normalized to Rps9 mRNA levels. Data are mean ± SD from three independent experiments.
(TIF)

**S4 Fig. Depletion of Nup358 impairs SG formation.** Antibody staining of Rin (red) or Lamin (green) of S2 cells treated with control dsRNA or Nup358 dsRNA in the presence or absence of arsenite. Hoechst staining is shown in blue. (B) Box plot of the number of Rin foci per cell. At least 30 cells were counted for each condition from two independent experiments. Data are mean ± SD. $p < 0.021(^*)$ by a one-way ANOVA (nonparametric) with a Bonferroni's post hoc-test.
(TIF)

**S5 Fig. Depletion of NXF1 impairs SG formation.** Antibody staining of Rin (red) of S2 cells treated with control dsRNA or NXF1 dsRNA (for 72 hrs) followed by one-hour treatment in the presence or absence of 500 μM sodium arsenite. Hoechst staining is shown in blue.
(TIF)

**S6 Fig. Expression of FLAG-Nup358.** Immunoblot of S2 cell lysates transfected with.
(TIF)

## Acknowledgments

We thank Eric Lecuyer for generously providing the Rin antibody. We acknowledge the UBC Life Science Institute core imaging facility for use of the Leica Sp5 microscope and Cellomics microscope. We thank Génome Québec for the support with RNA sequencing, Irvin Wason for helping with the gel filtration chromatography, and the Jan lab (Jodi Chien, Rachel DaSilva, Reid Warsaba and Christina Young) for discussions and critical reading of the paper.

## Author Contributions

**Conceptualization:** Eric Jan.

**Data curation:** Jibin Sadasivan, Marli Vlok, Xinying Wang, Eric Jan.

**Formal analysis:** Jibin Sadasivan, Marli Vlok, Eric Jan.

**Funding acquisition:** Eric Jan.

**Investigation:** Jibin Sadasivan, Xinying Wang, Arabinda Nayak, Raul Andino, Eric Jan.

**Methodology:** Jibin Sadasivan, Marli Vlok, Eric Jan.

**Project administration:** Eric Jan.

**Resources:** Raul Andino, Eric Jan.

**Supervision:** Eric Jan.

**Validation:** Jibin Sadasivan, Xinying Wang.

**Visualization:** Jibin Sadasivan.

**Writing – original draft:** Jibin Sadasivan, Eric Jan.

**Writing – review & editing:** Jibin Sadasivan, Marli Vlok, Xinying Wang, Arabinda Nayak, Raul Andino, Eric Jan.

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
