## [Decision Letter · Decision Letter 0]

1 Aug 2022

Dear Dr. Jan,

Thank you very much for submitting your manuscript "Targeting Nup358/RanBP2 by a viral protein disrupts stress granule formation" for consideration at PLOS Pathogens. As with all papers reviewed by the journal, your manuscript was reviewed by members of the editorial board and by several independent reviewers. In light of the reviews (below this email), we would like to invite the resubmission of a significantly-revised version that takes into account the reviewers' comments.

The three reviewers were enthusiastic regarding your submission - but each also has identified several areas in which the manuscript could be improved. Please address these in a revised manuscript

We cannot make any decision about publication until we have seen the revised manuscript and your response to the reviewers' comments. Your revised manuscript is also likely to be sent to reviewers for further evaluation.

Sincerely,

Jeffrey Wilusz

Guest Editor

PLOS Pathogens

Sara Cherry

Section Editor

PLOS Pathogens

Kasturi Haldar

Editor-in-Chief

PLOS Pathogens

orcid.org/0000-0001-5065-158X

Michael Malim

Editor-in-Chief

PLOS Pathogens

orcid.org/0000-0002-7699-2064

The three reviewers were enthusiastic regarding your submission - but each also has identified several areas in which the manuscript could be improved. Please address these in a revised manuscript

Reviewer's Responses to Questions

**Part I - Summary**

Reviewer #1: The manuscript by Sadasivan et al. explores stress granule (SG) modulation via the Cricket Paralysis Virus (CrPV) 1A protein. The CrPV-1A protein is the first nonstructural protein expressed in the viral life cycle (part of ORF1) and is a multifunctional protein implicated in host response modulation. Specifically, it has been shown to bind to and degrade Ago2 in an E3-ubiquitin ligase-dependent manner to block the antiviral RNA interference (RNAi) pathway and inhibit SG formation. Herein, the authors demonstrate that the R146 residue, but not the TALOS domain implicated in Ago-2 binding (explored through the F114 residue) was important for SG inhibition. Moreover, an intact BC-box domain, which couples to the ubiquitination machinery was required for this activity. The authors demonstrate that wild-type (WT) CrPV-1A but not R146-CrPV-1A localized to the nuclear membrane and led to the accumulation of polyA+ RNAs in the nucleus. Furthermore, they demonstrate that the nuclear pore protein, Nup358, is a CrPV-1A interacting protein, which is targeted for proteasomal degradation by WT-CrPV-1A. This suggests a model whereby Nup358 degradation leads to a decrease in nuclear export of polyA+ RNAs, a reduction in SG formation (which may or may not be connected to a reduction in Nup358 localization to these sites), and facilitates viral RNA replication/infection. The manuscript is well reasoned, well laid out, and the data convincing. The conclusions are well supported by the data; however, the discussion of some of the results could be improved by addressing some of the comments below. Overall, the article is of interest to the PloS Path audience and virology community.

Reviewer #2: Viruses employ numerous mechanisms to alter cellular pathways to favor infection. One such pathway is stress granule (SG) formation. These are RNP complexes that assemble in response to interrupted translation, often a virus-induced event. The Cricket Paralysis Virus (CrPV) 1A protein can bind to and degrade Ago2 via proteasomes to block antiviral RNAi and inhibit SG formation. The R146 residue of 1A is critical for the protein’s ability to inhibit SG formation and promote CrPV infection in Drosophila S2 cells and adult flies. In this work, the authors uncoupled 1A’s functions to ascertain how it blocks SG formation. Their results indicate that this 1A function requires its E3 ubiquitin-ligase binding domain and amino acid R146, but not the Ago2-binding TALOS element. R146 is the hub of numerous other functions, including: (i) localization of 1A to the nuclear membrane to promote nuclear enrichment of poly(A)+ RNA; (ii) transcriptome changes in infected cells; and (iii) targeting Nup358/RanBP2 degradation (which blocks SG formation).

The strengths of this work are that it is very interesting, tells an important story, and the data are excellent quality. Comments below are mostly minor, but the authors should address.

Reviewer #3: Dr. Jan’s lab has submitted an exciting manuscript that shows a new mechanism by which an RNA virus can modulate the formation of stress granules (SGs). SGs are ribonucleoprotein complexes that form in response to stress, such as a viral infection. Cellular mRNAs, RNA binding proteins and translation initiation complexes are stored within SGs. Many viruses circumvent the sequestration of translation initiation complexes in SGs by preventing the formation of stress granules either by degrading critical SG proteins such as G3BP1, or relocalizing different SG components. This manuscript describes a new mechanism whereby a viral nonstructural protein interferes with nuclear export of cellular mRNAs, and this function is regulated by a single amino acid namely R146.

Cricket paralysis virus (CrPV) nonstructural protein 1A modulates the RNAi pathway and SG formation. This protein is known to modulate the RNAi pathway by degrading Ago2 to prevent the targeting of small RNAs to the viral genome. This function is modulated by a loop region known as the TALOS domain and the investigators have previously shown that a single point mutation F114 affects this activity. The 1A protein also harbors a BC-box domain that recruits the ubiquitin ligase complex. Here the researchers investigate the mechanism by which 1A modulates SG formation. In particular, they focus on R146A mutant 1A protein, that was previously shown to attenuate virus infection which coincided with an increase in SG formation. Sadasivan and colleague use different approaches to dissect how 1A impairs SG formation. They first use a clever strategy of a mRNA reporter that expresses the CrPV 1A protein fused to a GFP, which when transfected into cells allows them to monitor cells expressing the mRNA (via GFP) and thus the effect of the 1A protein. This strategy also limits additional effects from virus infection. This approach elegantly shows that following arsenite treatment the 1A protein reduces the number of SG foci in a cell, however this restriction is countered by the R146A mutation. The authors show that the R146 mutation is specifically responsible for modulating SG formation as mutations in either the TALOS or BC-box domains had no effect. The 1A protein is found to localize at the nuclear periphery and poly(A)+ RNA accumulates in the nucleus. From RNA-seq studies they identify specific factors involved in nuclear export. While depletion of some of these factors does not affect expression of WT CrPV protein, the CrPV R146 mutant viruses show decreased VP2 protein levels following depletion of Rae1, Nup88 and Nup214. Notably, depletion of Nup358 did not affect viral protein and RNA of CrPV R146A. The authors show an increase in nuclear localization of poly(A)+ mRNA in cells depleted of NXF1 and Nup358, supporting the conclusion that 1A via Nup358 results in mRNA accumulation in the nucleus putatively by inhibiting nuclear export of mRNAs. The strengths of the manuscript include 1) a new mechanism showing how viruses can manipulate the formation of stress granules by interfering with proteins associated with the nuclear pore complex and hence nuclear export; 2) showing that RNA viruses that are typically thought to only manipulate processes in the cytoplasm have intimate interactions with the nuclear compartment, and 3) different experimental approaches to dissect the mechanism by which 1A R146A affects SG assembly.

Overall, this manuscript is well written, the experiments are well executed, and the authors have been meticulous in describing the data. The findings described in this manuscript are important to our understanding of how viruses subvert RNA granules in cell to promote virus infection.

**Part II – Major Issues: Key Experiments Required for Acceptance**

Reviewer #1: No major revisions requested, some additional information would be helpful, but not absolutely required (perhaps unless other reviewers also had the same queries), see below.

Reviewer #2: No major issues.

Reviewer #3: 1. The authors have data showing that 1A protein increases the amount of poly(A)+ RNA in the nucleus. These data are however based on quantification from immunofluorescence images. Since this is one of the data supporting the proposed mechanism, the investigators should include an orthogonal approach to validate these data. This might be undertaken by isolating nuclei and quantifying the amount of poly(A)+ RNA in the GFP-positive cells by RT-qPCR.

2. Figure 8D and lines 1094-1097: The authors show immunofluourescence data of poly(A)+ RNA and detection stress granules by visualizing Rin protein in foci. There are three concerns with these data.

i) The figure shows some poly(A)+ within nuclei. Here the authors state that in these cells Nup358 is depleted. However, the IFA does not show Nup358 expression/depletion, and a western blot of the depleted protein is not shown. The IFA should include visualization of Nup358 so that cells with increased poly(A)+ in the nucleus can be correctly correlated with Nup358 depletion. Sadasivan and colleagues should also include quantification of the extent of poly(A)+ nuclear localization.

ii) Depletion of Nup358 putatively retains mRNA in the nucleus which then limits the amount of RNA to be redistributed to SGs in the cytoplasm. The authors should show depletion of another similar protein (e.g., GP210) to demonstrate if this effect is a general response or specific to Nup358.

iii) The authors should include an experiment that rescues the effect by over expressing a RNAi-resistant Nup358.

3. The model figure suggests an interaction between 1A protein and Nup358. While the authors have been cautious to state that this might be a direct or indirect effect, showing such an interaction by co-IP would further support the proposed model by which 1A is affecting SG assembly. Alternatively, Sadasivan and colleagues could perform the same Z-stack microscopy analysis of 1A and lamin in cells depleted of Nup358. The prediction would be that 1A was no longer localized at the nuclear periphery.

4. Figure 7: from the figure legend it seems that the data should show images from mock- CrPV-WT and CrPV-R146A virus infections. The image panel does not represent this description. The figure should also include quantification of the amount of poly(A)+ in the nucleus.

5. Line 385: “Knockdown of Rae1, mTor, Nup88 or Nup214 did not restore VP2 expression under mutant CrPV (R146A) virus infection compared to wild-type infection” The authors do not specifically state the number of times that the quantitation of the western blot of VP2 was performed but looking at the error bars in Figure 8B, it seems that mTor does restore infection of the CrPV (R146A) virus. Undertaking RT-qPCR to determine the relative amount of viral RNA in these knockdown cells would clarify this.

6. Figure 8: The authors use RNAi to deplete a subset of proteins involved in mRNA export yet no experiments are provided to validate the depletion of the proteins or at least depletion of transcript levels.

7. The formation of SGs formation is assayed following PatA treatment in cells that by western blot show efficient depletion of Nup358. However, CrPV infection only causes a 50% decrease in Nup358. It would be good to correlate the extent of Nup358 turnover with the extent of SG formation/inhibition?

8. Why does MG132 decrease VP2 levels? This is a curious observation, and it would be useful if the authors discussed this further in the discussion section.

9. In Figure 3 Sadavisan et al., show that R146A affects the proteolytic processing of the polyprotein. It is disappointing that the authors only briefly mention this observation or the effect of this defect in the discussion. It would seem that decreased 2A proteolytic processing could have important effects on SG formation, nuclear accumulation of poly(A)+ RNA or degradation of Nup358.

**Part III – Minor Issues: Editorial and Data Presentation Modifications**

Reviewer #1: (No Response)

Reviewer #2: 1. Fig. 3A. page 13, line 257: the text states the individual CrPV-1A proteins were detected at similar levels after transfection. However, their levels seem to vary quite a bit, e.g., L17A is significantly lower than F114A, R146A, and WT. Could the authors clarify their statement?

2. Fig. 3A, page 13, line 261: the authors note that a longer exposure reveals an unprocessed 1A-2A-GFP protein for the 1A (R146A) mutant. However, this must be a different blot since there is a gap between GFP lanes 7 and 8 in the lower exposure that is missing from the longer exposure. Please clarify.

3. Fig. 7A: the labels are reversed for the Hoechst and Poly A images. Additionally, the Hoechst image for the GFP dsRNA control is very faint compared to the NXF1 dsRNA image.

4. Fig. 7B: authors should show efficiency of protein reduction by Western blot following dsRNA-mediated knockdowns.

5. Fig. 8A: the authors should show efficiency of protein reduction by Western blot following dsRNA-mediated knockdowns.

6. Fig. 8C, page 19, line 395: if there is decreased viral RNA with GP210 knockdown for R146A (panel C), could the authors explain the comparable levels of VP2 protein upon GP210 knockdown with WT and R146A (panel B).

Reviewer #3: 1. Line 121: Misalignment of formatting of references “termed antiviral SGs (avSGs)6776,”

2. Line 223: period missing at the end of the sentence

3. Figure 2: It would be good to include data from the WT in Figure 2 so that the rescue effect of mutations within the BC box can be more easily compared

4. Line 1097: A period is missing at the end of the sentence. “The arrows show Nup358 knockdown cells.”

PLOS authors have the option to publish the peer review history of their article (what does this mean?). If published, this will include your full peer review and any attached files.

Reviewer #1: No

Reviewer #2: No

Reviewer #3: No
---

## [Decision Letter · Decision Letter 1]

17 Nov 2022

Dear Dr. Jan,

We are pleased to inform you that your manuscript 'Targeting Nup358/RanBP2 by a viral protein disrupts stress granule formation' has been provisionally accepted for publication in PLOS Pathogens.

Best regards,

Jeffrey Wilusz

Guest Editor

PLOS Pathogens

Sara Cherry

Section Editor

PLOS Pathogens

Kasturi Haldar

Editor-in-Chief

PLOS Pathogens

orcid.org/0000-0001-5065-158X

Michael Malim

Editor-in-Chief

PLOS Pathogens

orcid.org/0000-0002-7699-2064

I wish to thank you, Jibin, Marli, Xinying, Arabinda and Raul for effectively responding to all of the points raised in the initial round of review.

Reviewer Comments (if any, and for reference):

Reviewer's Responses to Questions

**Part I - Summary**

Reviewer #3: This is an exciting, revised manuscript that shows that a viral nonstructural protein helps promote infection by interfering with the export of cellular mRNAs from the nucleus, and this affects the formation of stress granules. This proviral activity is directed by a single amino acid namely R146 in the 1A protein of CrPV. These findings provide new insight as to how viruses subvert cellular RNA granules to promote virus infection. Dr. Jan and co-authors have been responsive to the suggested changes, and where possible have included additional data. No additional changes are needed.

**Part II – Major Issues: Key Experiments Required for Acceptance**

Reviewer #3: (No Response)

**Part III – Minor Issues: Editorial and Data Presentation Modifications**

Reviewer #3: (No Response)

PLOS authors have the option to publish the peer review history of their article (what does this mean?). If published, this will include your full peer review and any attached files.

Reviewer #3: No

---

## [Editor Report · Acceptance letter]

28 Nov 2022

Dear Dr. Jan,

We are delighted to inform you that your manuscript, "Targeting Nup358/RanBP2 by a viral protein disrupts stress granule formation," has been formally accepted for publication in PLOS Pathogens.

Best regards,

Kasturi Haldar

Editor-in-Chief

PLOS Pathogens

orcid.org/0000-0001-5065-158X

Michael Malim

Editor-in-Chief

PLOS Pathogens

orcid.org/0000-0002-7699-2064